# FUSE: FULL-SPECTRUM UNLEARNABLE EXAMPLES VIA SPECTRAL EQUALIZATION

## ABSTRACT

Unlearnable examples (UEs) protect training data by injecting imperceptible per-turbations so that models fail to extract exploitable representations. In this paper, we reveal that existing UEs exhibit a critical failure once low-pass filtering is applied, indicating that the effective perturbation signals for unlearnability con-centrate predominantly in high frequencies. Hence, we argue that reliable UEs should remain effective across the full spectrum. To this end, we propose **F**ull-spectrum **U**nlearnable examples via **S**pectral **E**qualization (**FUSE**), which aims to generate spectrum-agnostic perturbations by equalizing the contributions from different bands and enforcing cross-band consistency. Specifically, FUSE adopts a Random Spectral Masking (RSM) strategy during generator training, which randomly removes a contiguous frequency band, forcing the remaining bands to maintain unlearnability. In addition, FUSE further integrates Cross-Band Guid-ance (CBG), which enforces mutual consistency between high- and low-frequency components, thereby further enhancing low-frequency unlearnability and regu-lating high-frequency perturbations to preserve the semantic fidelity of images. Extensive experiments across multiple datasets, architectures, and spectral filtering demonstrate the strong protection achieved by FUSE.

## 1 INTRODUCTION

With the rapid growth of social media and online sharing platforms, an increasing number of individuals are sharing images from their daily lives. While this enhances connectivity and information exchange, it also raises the risk that personal images may be repurposed to train commercial models without their owners' consent. As deep learning methods (Deng et al., 2009; Brown, 2020; Liu et al., 2023) increasingly rely on massive datasets (Russakovsky et al., 2015) to achieve stronger performance, the potential for privacy breaches has become a pressing concern. To mitigate this risk, the task of generating Unlearnable Examples (UEs) has been proposed, where imperceptible perturbations are added to images such that models fail to extract correct semantic information from them (Huang et al., 2021; Fu et al., 2022; Ren et al., 2022).

Despite recent progress, we show that existing UE defenses can be easily circumvented. Specifically, our empirical analysis demonstrates that simply applying a low-pass filter to protected data can substantially undermine their unlearnability (Figure 1), indicating that the generated perturbations depend predominantly on high-frequency components. Once these are suppressed, models can still extract semantic information from the residual low-frequency content, ultimately leading to privacy leakage. These results expose a fundamental vulnerability of current approaches and suggest that their protective capability has been overestimated.

In this paper, we argue that a key requirement for reliable UEs is to remain effective under spectral filtering, which in turn demands that the perturbations themselves function across the entire spectrum so that no semantic information can be extracted from any particular band. Otherwise, models may still exploit residual cues in bands left unaffected by perturbations. If perturbations act only on high-frequency regions, low-frequency structures such as object shapes remain informative and continue to support learning. Conversely, when perturbations concentrate on low frequencies, fine-grained details in the high-frequency range can still be exploited. This insight motivates a spectrum-agnostic design that regulates perturbation effects toward a balanced distribution across the spectrum, thereby preserving unlearnability even when certain frequency bands are suppressed by filtering.

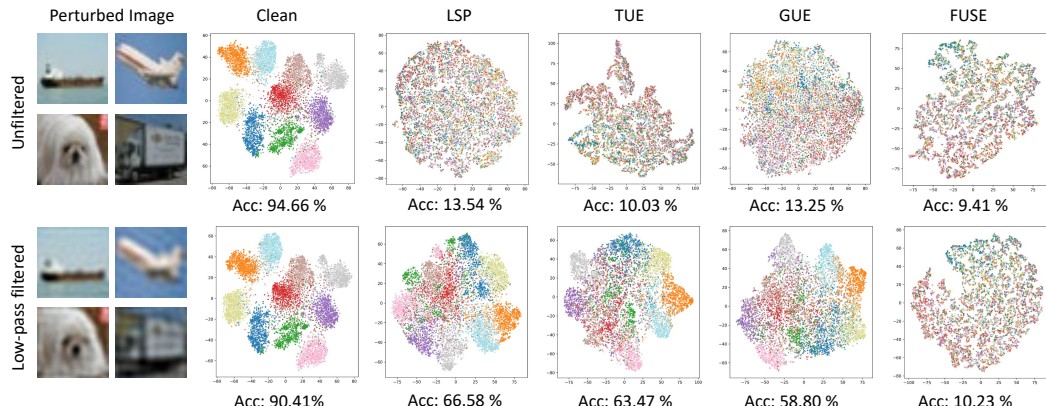

Figure 1: t-SNE visualization of the last-layer features, where classifiers are trained on the perturbed CIFAR-10 dataset and evaluated on the clean test set. Visualizations are shown for different UE methods under the unfiltered setting (top) and when a low-pass filter is applied to the perturbed images before training (bottom). While existing approaches (e.g., LSP (Yu et al., 2022), TUE (Ren et al., 2022), GUE (Liu et al., 2024a)) suffer severe performance degradation once high-frequency components are suppressed, FUSE remains effective and maintains features close to random guessing.

To this end, we propose **F**ull-spectrum **U**nlearnable examples via **S**pectral **E**qualization (**FUSE**), a novel method explicitly designed to generate spectrum-agnostic perturbations that maintain unlearnability even under spectral filtering. Specifically, FUSE incorporates two complementary components. (i) *Random Spectral Masking* (RSM), introduces randomized spectral band suppression during training to simulate diverse distortions. This mechanism ensures perturbations to remain effective regardless of which frequencies are suppressed. (ii) *Cross-Band Guidance* (CBG), further boosts unlearnability and utility of UEs by mutual guidance between low- and high-frequency signals, enhancing their consistency. This mechanism allows low-frequency components to inherit unlearnability from high frequencies and high-frequency components to preserve semantic fidelity from low frequencies, ensuring that both collectively retain complementary strengths. Together, RSM and CBG enable FUSE to generate spectrum-agnostic perturbations that effectively degrade semantic learnability, thereby offering strong and reliable privacy protection across diverse scenarios.

Our main contributions are summarized as follows:

- We identify a critical vulnerability of existing unlearnable examples, showing that their effectiveness collapses under low-pass filtering.

- We propose FUSE, a novel spectrum-agnostic framework that distributes perturbation effects across the entire frequency range, ensuring that unlearnability is preserved even when specific bands are suppressed by filtering.

- Through extensive experiments across datasets, architectures, and filtering scenarios, we demonstrate that FUSE consistently outperforms prior approaches, achieving strong unlearnability while maintaining imperceptibility.

## 2    RELATED WORK

Unlearnable examples (UEs) have recently emerged as an effective paradigm for safeguarding data privacy by adding carefully crafted perturbations to training samples. The key idea is to render data unusable for model training while preserving visual imperceptibility, thereby preventing unauthorized exploitation of sensitive information. Models trained on UEs typically memorize spurious patterns introduced by perturbations rather than learning genuine semantic features, which results in severe performance degradation when evaluated on clean test data (Huang et al., 2021; Fu et al., 2022).

Existing approaches to generating UEs generally focus on designing perturbations that mislead models into relying on spurious patterns rather than true semantic information. Some works achieve this by embedding signals that dominate the learning process and obscure meaningful features (Huang et al., 2021; Liu et al., 2024a; Ren et al., 2022). Others introduce perturbations that deliberately push

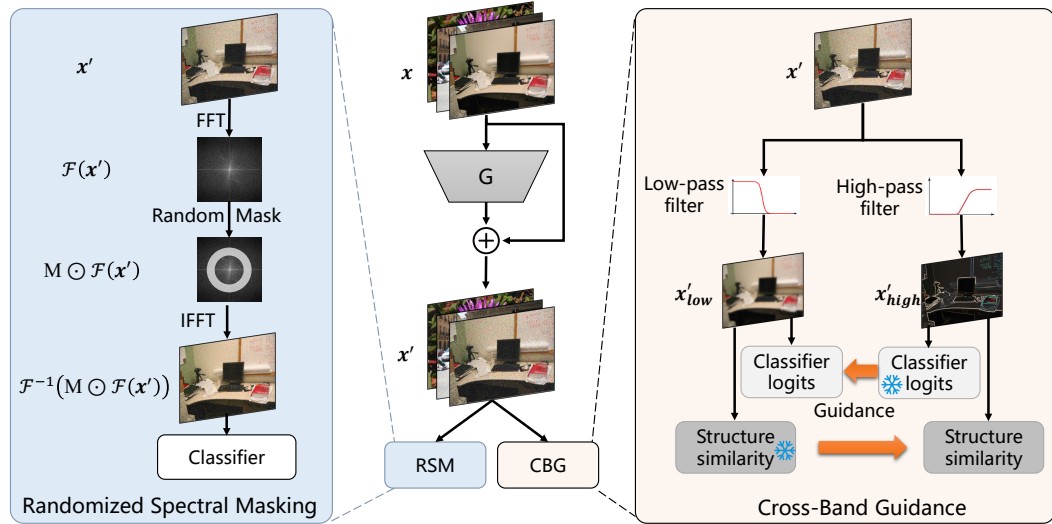

Figure 2: Overview of the proposed Full-spectrum Unlearnable Examples via Spectral Equalization (FUSE) framework. FUSE consists of two complementary components: (i) *Random Spectral Masking* (RSM), which applies randomized band suppression in the Fourier domain to prevent perturbations from collapsing into specific frequencies, and (ii) *Cross-Band Guidance* (CBG), which enforces mutual consistency between low- and high-frequency signals by aligning structural similarity and guiding classifier predictions. Together, RSM and CBG distribute perturbation effects across the full spectrum and ensure that unlearnability is preserved even under spectral filtering.

samples toward incorrect decision regions, causing classifiers to associate images with misleading labels (Chen et al., 2024; Zhang et al., 2023; Liu et al., 2024b). More recently, model-independent strategies have emerged, where perturbations are constructed without the use of surrogate models, for example by leveraging random convolutions or enforcing linear separability across classes (Yu et al., 2022; Sadasivan et al., 2023; Sandoval-Segura et al., 2022). While these methods provide effective protection in standard training scenarios, they generally overlook the impact of frequency characteristics, causing their unlearnability to diminish substantially under low-pass filtering. Beyond UEs, several recent works (Li et al., 2024) underscore the importance of frequency-domain analysis, complementing our motivation for developing a spectrum-agnostic UE method.

## 3 METHOD

In this section, we present Full-spectrum Unlearnable Examples via Spectral Equalization (FUSE), a spectrum-agnostic framework designed to preserve unlearnability under diverse spectral filtering, as illustrated in Figure 2. Specifically, we adopt a *Random Spectral Masking* (RSM) strategy, which randomly removes a contiguous frequency band, thereby forcing the remaining bands to sustain the unlearnable effect. In addition, we introduce *Cross-Band Guidance* (CBG), which enforces mutual consistency between low- and high-frequency signals, strengthening low-frequency unlearnability while regulating high-frequency perturbations to preserve semantic fidelity.

### 3.1 PROBLEM SETUP

We formulate the problem in the context of image classification with deep neural networks (DNNs). Let $(\boldsymbol{x}, y)$ be a labeled example, where $\boldsymbol{x} \in \mathcal{X}$ is an input image and $y \in \mathcal{Y} = \{1, \ldots, K\}$ is its class label. We denote the clean training and test datasets as $\mathcal{D}_c$ and $\mathcal{D}_t$, respectively. A perturbation generator $\mathcal{G}_\psi$ is applied to $\mathcal{D}_c$ to produce an unlearnable dataset $\mathcal{D}_u$ by adding generated perturbations $\boldsymbol{x}' = \boldsymbol{x} + \boldsymbol{\delta}$, where $\boldsymbol{\delta} = \mathcal{G}_\psi(\boldsymbol{x})$ and $\psi$ denotes the learnable parameters of the generator. To capture scenarios where images are subject to spectral filtering, we additionally consider the case where training data is processed with a low-pass filter. In this setting, a classifier is trained on low-pass filtered unlearnable data, and the effectiveness of $\boldsymbol{\delta}$ is evaluated by the accuracy degradation of classifier on the clean test set $\mathcal{D}_t$. The objective is to design perturbations that consistently prevent

the model from learning useful semantic information, even when the perturbed data is modified by low-pass filtering.

## 3.2 Randomized Spectral Masking

To ensure that perturbations remain effective under spectral filtering, we propose Randomized Spectral Masking. Specifically, RSM enforces spectral invariance by combining randomized frequency dropout, which prevents reliance on particular bands, with entropy-based regularization that encourages dispersion of perturbation energy across the spectrum. In this way, UEs are discouraged from concentrating in narrow spectral regions and remain effective under diverse filtering conditions.

To avoid perturbations collapsing onto a narrow spectral region, we introduce a stochastic frequency dropout mechanism that, at each training step, randomly discards a contiguous portion of the frequency domain. This design ensures that the remaining frequency components can still achieve the unlearnable objective, thereby compelling perturbations to distribute their influence across the spectrum. By applying different random masks at every step, the model learns to generate spectrum-agnostic UEs that remain effective regardless of which frequencies are suppressed.

Given an input $\mathbf{z}$, the random mask operator is defined as

$$\mathcal{T}_{\text{spec}}(\mathbf{z}) = \mathcal{F}^{-1}(\mathbf{M} \odot \mathcal{F}(\mathbf{z})), \tag{1}$$

where $\mathcal{F}$ denotes the Fourier transform and $\mathbf{M} \in \{0, 1\}^{H \times W}$ is a binary mask that zeros out a randomly sampled contiguous spectral band at each training step.

By training with randomized frequency suppression, the perturbation is required to preserve its unlearnable effect regardless of which frequency subset remains. This design prevents over-reliance on a specific spectral region and instead enforces a broader distribution of perturbation influence. The associated invariance loss is defined as

$$\mathcal{L}_{\text{inv-spec}} = \mathcal{L}_{\text{CE}}\left(f_\theta\left(\mathcal{T}_{\text{spec}}^{(k)}(\boldsymbol{x} + \boldsymbol{\delta})\right), y\right), \tag{2}$$

where $f_\theta$ is the classifier and $\mathcal{T}_{\text{spec}}^{(k)}$ denotes the operator with an independently random mask at step $k$. By optimizing this loss, the perturbation is compelled to remain unlearnable even when part of the spectrum is dropped, thereby enhancing its effectiveness across the entire frequency range.

Randomized masking ensures invariance under diverse filtering, but it does not necessarily prevent perturbation energy from collapsing into narrow spectral regions. To address this issue, we introduce a regularizer that promotes a more balanced energy allocation by maximizing the entropy of the normalized power spectrum. Formally, We denote $\mathcal{F}[\delta]$ as the 2D FFT of the perturbation $\delta$. We apply fftshift to convert the raw FFT output into a centered spectrum representation, where the zero-frequency component lies at the center: $\hat{\boldsymbol{\delta}} = \text{fftshift}(\mathcal{F}[\boldsymbol{\delta}])$. The normalized spectral energy distribution is:

$$P(u, v) = \frac{|\hat{\boldsymbol{\delta}}(u, v)|^2}{\sum_{u,v} |\hat{\boldsymbol{\delta}}(u, v)|^2}, \tag{3}$$

where $P(u, v)$ is interpreted as the probability mass assigned to frequency coordinate $(u, v)$. The spectral entropy is then defined as

$$\mathcal{H}(\boldsymbol{\delta}) = -\sum_{u,v} P(u, v) \log P(u, v). \tag{4}$$

A larger value of $\mathcal{H}(\boldsymbol{\delta})$ reflects a more uniform distribution of perturbation energy across frequencies, whereas a smaller value indicates that energy is concentrated within a limited region. Importantly, this regularizer does not enforce a perfectly uniform distribution (which would lead to white-noise perturbations), but rather mitigates excessive spectral bias while preserving the unlearnable objective. The complete RSM objective integrates the invariance and entropy terms:

$$\mathcal{L}_{\text{RSM}} = \mathcal{L}_{\text{inv-spec}} - \mathcal{H}(\boldsymbol{\delta}). \tag{5}$$

This formulation promotes perturbations that remain unlearnable even under strong spectral filtering, thereby mitigating the vulnerability of prior approaches whose unlearnability is largely confined to high-frequency components, and thus fails to generalize under low-pass filtering.

## 3.3 CROSS-BAND GUIDANCE

While randomized spectral masking distributes perturbations across the spectrum, it does not explicitly constrain the interaction between low- and high-frequency components. In natural images, low-frequency components primarily capture global semantics such as object shape, whereas high-frequency components represent local details and texture. To prevent models from exploiting either band independently, we propose Cross-Band Guidance, which imposes mutual constraints between the two bands to ensure their complementary roles are jointly preserved.

Given a clean image $\boldsymbol{x}$ and its perturbed counterpart $\boldsymbol{x}'$, we define the frequency split radius $r_c \in [0, 1]$ in the normalized Fourier domain so that it remains independent of the input resolution. For an image of spatial size $H \times W$, the normalized radial frequency at location $(i, j)$ is computed as:

$$r(i, j) = \sqrt{\left(\frac{i - H/2}{H/2}\right)^2 + \left(\frac{j - W/2}{W/2}\right)^2}. \tag{6}$$

We construct binary circular masks for low- and high-frequency regions:

$$M_{\text{low}}(i, j) = \mathbb{1}\left[r(i, j) \le r_c\right], \qquad M_{\text{high}} = 1 - M_{\text{low}}. \tag{7}$$

The decomposed components are obtained by masking the Fourier transform of the image:

$$\begin{aligned} \boldsymbol{x}_{\text{low}}, \boldsymbol{x}_{\text{high}} &= \mathcal{F}^{-1}\big(M_{\text{low}} \odot \mathcal{F}(\boldsymbol{x})\big), \ \mathcal{F}^{-1}\big(M_{\text{high}} \odot \mathcal{F}(\boldsymbol{x})\big), \\ \boldsymbol{x}'_{\text{low}}, \boldsymbol{x}'_{\text{high}} &= \mathcal{F}^{-1}\big(M_{\text{low}} \odot \mathcal{F}(\boldsymbol{x}')\big), \ \mathcal{F}^{-1}\big(M_{\text{high}} \odot \mathcal{F}(\boldsymbol{x}')\big). \end{aligned} \tag{8}$$

A small $r_c$ corresponds to coarse global structures, whereas a larger $r_c$ incorporates finer details. By explicitly modeling this decomposition, CBG establishes two complementary pathways that capture global semantics and fine details, and their interaction is critical for achieving effective unlearnability. We evaluate the impact of $r_c$ in the ablation study.

To couple the two bands, we introduce a semantic guidance mechanism that transfers the stronger unlearnability of high-frequency perturbations to the low-frequency branch. High-frequency perturbations naturally induce stronger confusion in the classifier, producing unstable or incorrect predictions. We therefore treat the high-frequency prediction as a soft target and use the following joint formulation:

$$\hat{y}_{\text{high}} = f_\theta(\boldsymbol{x}'_{\text{high}}), \quad \mathcal{L}_{\text{guide}} = \mathcal{L}_{\text{CE}}(f_\theta(\boldsymbol{x}'_{\text{low}}), \hat{y}_{\text{high}}). \tag{9}$$

This design encourages the low-frequency component to remain calibrated with the semantics emphasized by the high-frequency band, rather than injecting arbitrary distortions. In effect, the low-frequency pathway provides stable semantic cues, while the high-frequency pathway is constrained to reinforce these cues. This prevents the model from exploiting high-frequency artifacts as an alternative shortcut and ensures that the perturbations across different bands contribute coherently to the unlearnability objective.

Since low-frequency bands encode global semantics that remain relatively stable under perturbations, whereas high-frequency bands are more vulnerable to distortions, the low-frequency branch serves as a reliable reference to regularize high-frequency perturbations. We realize this guidance by measuring a hybrid similarity that integrates feature-level and perceptual cues. Let $\phi(\cdot)$ denote the feature extractor of the surrogate model $f_\theta$, which is kept frozen during training, and let $\cos(\cdot, \cdot)$ denote the cosine similarity. The hybrid similarity for each band is then defined as

$$\text{sim}(\boldsymbol{a}, \boldsymbol{b}) = \cos\big(\phi(\boldsymbol{a}), \phi(\boldsymbol{b})\big) + \text{SSIM}(\boldsymbol{a}, \boldsymbol{b}). \tag{10}$$

For the low- and high-frequency components, the band-wise similarities are

$$s_{\text{low}} = \text{sim}(\boldsymbol{x}_{\text{low}}, \boldsymbol{x}'_{\text{low}}), \qquad s_{\text{high}} = \text{sim}(\boldsymbol{x}_{\text{high}}, \boldsymbol{x}'_{\text{high}}). \tag{11}$$

To let the low-frequency branch guide the high-frequency branch, we match the two similarities via a consistency loss:

$$\mathcal{L}_{\text{struct}} = \text{MSE}(s_{\text{high}}, \text{sg}[\, s_{\text{low}} \,]), \tag{12}$$

where $\text{sg}[\cdot]$ denotes the stop-gradient operator, ensuring that the low-frequency similarity serves as a fixed target without receiving gradients from $\mathcal{L}_{\text{struct}}$. This consistency regularizer discourages

perturbations from distorting one band independently and encourages balanced interaction between low and high frequencies.

By jointly enforcing semantic guidance and structural alignment in both feature and perceptual domains, CBG balances low- and high-frequency perturbations to enhance unlearnability without compromising perceptual plausibility. Finally, the complete CBG objective combines semantic guidance with structural alignment:

$$\mathcal{L}_{\text{CBG}} = \mathcal{L}_{\text{guide}} + \lambda \cdot \mathcal{L}_{\text{struct}}, \tag{13}$$

where $\lambda$ balances the trade-off between enforcing semantic guidance and preserving structural fidelity. Through the integration of frequency decomposition, semantic guidance, and structural alignment, CBG strengthens low-frequency unlearnability while constraining high-frequency perturbations to maintain semantic fidelity, preventing the model from relying excessively on any single band.

### 3.4 Optimization Objective

In summary, our framework integrates two complementary components: *Randomized Spectral Masking*, which enforces spectrum-agnostic unlearnability by preventing perturbations from collapsing onto specific frequency bands, and *Cross-Band Guidance*, which strengthens low-frequency unlearnability while constraining high-frequency perturbations to preserve semantic fidelity. Together, these designs yield perturbations that remain effective across spectral filtering while maintaining visual coherence. We delineate the complete training procedure in Algorithm 1. Formally, the overall optimization is:

$$\arg\min_{\theta} \ \mathbb{E}_{(\boldsymbol{x},y) \sim \mathcal{D}_c} \left[ \min_{\psi} \mathcal{L}_{\text{full}} \big( f_\theta(\boldsymbol{x} + \mathcal{G}_\psi(\boldsymbol{\delta})), y \big) \right], \quad \text{s.t. } \|\boldsymbol{\delta}\|_\infty \leq \epsilon, \tag{14}$$

where $\epsilon = 8/255$ denotes the perturbation budget. The final training objective is defined as:

$$\mathcal{L}_{\text{full}} = \mathcal{L}_{\text{RSM}} + \mathcal{L}_{\text{CBG}}. \tag{15}$$

---

**Algorithm 1** Full-Spectrum Unlearnable Noise Generation

---

**Input:** Surrogate model $f_\theta$, random mask operator $\mathcal{T}_{\text{spec}}$, frequency splitter (FFT) $\mathcal{F}$, hybrid similarity sim, training data $(\boldsymbol{x}, y) \in \mathcal{D}_c$, balance coefficient $\lambda$, frequency split radius $r_c$, first loop training steps $T$, training epochs $E$;

**Output:** Full-Spectrum perturbation generator $\mathcal{G}_\psi$;

1: **for** $e = 1$ to $E$ **do**
2:     **for** $t = 1$ to $T$ **do**
3:         $i, \boldsymbol{x}_i, y_i = \text{Next}(\boldsymbol{x}, y)$;
4:         Input $\boldsymbol{x}_i$ to $\mathcal{G}_\psi$ and get $\boldsymbol{\delta} = \mathcal{G}_\psi(\boldsymbol{x}_i)$;
5:         Input $\boldsymbol{x}_i$ and $\boldsymbol{\delta}$ to $f_\theta$ to calculate $\mathcal{L}_{\text{CE}}$;
6:         Update $f_\theta$ by minimizing $\mathcal{L}_{\text{CE}}$;
7:     **end for**
8:     **for** $\boldsymbol{x}_j, y_j$ in $\boldsymbol{x}, y$ **do**
9:         Input $\boldsymbol{x}_j$ to $\mathcal{G}_\psi$ and get $\boldsymbol{\delta} = \mathcal{G}_\psi(\boldsymbol{x}_j)$;
10:        Obtain unlearnable example $\boldsymbol{x}'_j = \boldsymbol{x}_j + \boldsymbol{\delta}$;
11:        Input $\boldsymbol{x}'_j$ to $\mathcal{T}_{\text{spec}}$ and $f_\theta$ and calculate invariance loss $\mathcal{L}_{\text{inv-spec}}$;
12:        Calculate spectrum entropy $\mathcal{H}(\boldsymbol{\delta})$;
13:        Calculate $M_{\text{low}}(u, v) = \mathbb{1}[r(u, v) \leq r_c]$ and $M_{\text{high}} = 1 - M_{\text{low}}$;
14:        Input $\boldsymbol{x}_j$ and $\boldsymbol{x}'_j$ to $M_{\text{low}}$ and $M_{\text{high}}$ and calculate $\boldsymbol{x}_{j,\text{low}}, \boldsymbol{x}_{j,\text{high}}, \boldsymbol{x}'_{j,\text{low}}, \boldsymbol{x}'_{j,\text{high}}$;
15:        Use $f_\theta(x'_{j,\text{high}})$ as soft target and $f_\theta(x'_{j,\text{low}})$ to compute $\mathcal{L}_{\text{guide}}$;
16:        Input $\boldsymbol{x}_{j,\text{low}}, \boldsymbol{x}_{j,\text{high}}, \boldsymbol{x}'_{j,\text{low}}, \boldsymbol{x}'_{j,\text{high}}$, to sim and calculate $\mathcal{L}_{\text{struct}}$;
17:        $\mathcal{L}_{\text{full}} = \mathcal{L}_{\text{inv-spec}} - \mathcal{H}(\boldsymbol{\delta}) + \mathcal{L}_{\text{guide}} + \lambda\mathcal{L}_{\text{struct}}$;
18:        Update $\mathcal{G}_\psi$ by minimizing $\mathcal{L}_{\text{full}}$;
19:     **end for**
20: **end for**
21: **return** Full-Spectrum perturbation generator $\mathcal{G}_\psi$

---

# 4 EXPERIMENTS

In this section, we conduct extensive experiments to evaluate the effectiveness of our proposed FUSE framework across multiple datasets and architectures. We begin by examining its unlearnability under low-pass and unfiltered settings to validate our design motivation. We then compare FUSE with representative UE baselines under varying cutoff levels to assess effectiveness across different degrees of frequency suppression. We also present ablation studies, including hyperparameter analysis, to isolate the individual contributions of Random Spectral Masking and Cross-Band Guidance. Finally, we evaluate our method against several defense strategies.

## 4.1 EXPERIMENTAL SETTINGS

**Datasets and Models.** We evaluate FUSE on CIFAR-10 (Krizhevsky et al., 2009), CIFAR-100 (Krizhevsky et al., 2009), SVHN (Netzer et al., 2011). We use ResNet-18 (He et al., 2016), ResNet-50 (He et al., 2016), VGG-11 (Simonyan & Zisserman, 2014), DenseNet-121 (Huang et al., 2017), and ViT (Dosovitskiy et al., 2020) as the surrogate classifier and target model both in training and testing. We utilize the classification accuracy on the clean test set as the evaluation metric, where lower accuracy indicates stronger unlearnability and protectiveness. Unless otherwise specified, low-pass filtering is implemented in the Fourier domain, where the cutoff is defined by a normalized Fourier radius of 0.5 relative to the image size.

**Baseline.** We choose five UE methods as baselines, which are Error-Minimizing Noise (EMN) (Huang et al., 2021), Linearly-Separable Perturbations (LSP) (Yu et al., 2022), Transferable Unlearnable Examples (TUE) (Ren et al., 2022), Game Unlearnable Example (GUE) (Liu et al., 2024a), Provably Unlearnable Examples (PUE) (Wang et al., 2024).

**Implementation Details.** Our model is developed with the PyTorch framework and trained on a single NVIDIA L40S GPU. We employ the Adam optimizer (Kingma & Ba, 2015) with an initial learning rate of 0.001. We set the first loop training step $T$ to 10 in all experiments. We train our model for 50 epochs on CIFAR-10 and SVHN, and 60 epochs on CIFAR-100. We fix the frequency split radius in Eq. 7 to $r_c = 0.5$ and the balance coefficient in Eq. 13 to $\lambda = 0.5$ throughout all experiments. To ensure imperceptibility, we set the perturbation bound $\epsilon$ to $8/255$ in Eq. 14 following prior works Huang et al. (2021); Liu et al. (2024a); Chen et al. (2024). Results are averaged over five runs with different random seeds.

## 4.2 MAIN RESULTS

**Low-pass Filtering across Backbones.** We evaluate the performance of representative UE methods under low-pass filtering across multiple backbone architectures. This setting explicitly tests whether perturbations remain effective when high-frequency information is progressively suppressed. Results on CIFAR-10, CIFAR-100, and SVHN allow us to examine performance across datasets with varying semantic complexity and visual diversity. As shown in Table 1, existing methods exhibit substantial recovery in test accuracy once low-pass filters are applied. In particular, on CIFAR-10 with ResNet-18, LSP and TUE yield test accuracies of $54.16\%$ and $52.16\%$, respectively, indicating a sharp degradation of unlearnability. In contrast, our proposed FUSE achieves only $10.13\%$. Similar trends are observed on CIFAR-100 and SVHN, where FUSE achieves below $5\%$ accuracy on CIFAR-100 and around $10\%$ on SVHN across architectures. These results demonstrate that FUSE produces spectrum-agnostic perturbations that remain effective even when high-frequency components are removed, thereby overcoming the vulnerability of prior methods to low-pass filtering.

**Unfiltered Evaluation across Backbones.** We further evaluate unlearnable examples on unfiltered images to assess their effectiveness without spectral filtering. As shown in Table 2, FUSE achieves the lowest test accuracy across most dataset–architecture pairs, confirming that its perturbations remain effective even without spectral distortions. On CIFAR-10 with ResNet-18, FUSE reduces accuracy to $9.41\%$, substantially lower than the UE baseline LSP at $13.54\%$. On SVHN with ViT, FUSE reaches $6.55\%$, approaching the random-guessing level for a ten-class task. Even in the rare cases where FUSE is not the best performer, its performance remains very close, indicating negligible learnability. These findings demonstrate that FUSE is not confined to manipulating low-frequency information; rather, it retains strong unlearnability across the entire spectrum. This highlights that our framework produces perturbations that are intrinsically effective under both low-pass and unfiltered settings,

Table 1: Test accuracy (%) ↓ of UE methods across different backbones under low-pass filtering.

| Dataset | Backbone | Clean | EMN | LSP | TUE | GUE | PUE | FUSE |
|---------|----------|-------|-----|-----|-----|-----|-----|------|
| CIFAR-10 | ResNet-18 | 90.41 | 23.56 | 54.16 | 52.16 | 41.97 | 23.60 | **10.13** |
| | ResNet-50 | 90.49 | 21.03 | 52.82 | 57.01 | 43.02 | 19.26 | **10.34** |
| | VGG-11 | 85.01 | 29.91 | 57.16 | 74.74 | 44.78 | 30.44 | **10.20** |
| | DenseNet-121 | 91.10 | 24.51 | 55.12 | 58.26 | 42.98 | 23.96 | **10.47** |
| | ViT | 66.16 | 29.63 | 51.99 | 60.58 | 45.08 | 27.01 | **10.22** |
| CIFAR-100 | ResNet-18 | 63.20 | 28.64 | 28.25 | 58.64 | 37.54 | 56.23 | **4.45** |
| | ResNet-50 | 64.11 | 28.77 | 25.31 | 54.95 | 49.64 | 63.05 | **3.26** |
| | VGG-11 | 56.69 | 28.91 | 20.03 | 55.79 | 48.46 | 56.18 | **4.21** |
| | DenseNet-121 | 65.55 | 29.29 | 26.93 | 52.07 | 50.73 | 58.72 | **1.49** |
| | ViT | 37.87 | 29.56 | 21.61 | 37.26 | 33.20 | 37.39 | **1.53** |
| SVHN | ResNet-18 | 95.93 | 18.25 | 22.79 | 22.39 | 20.28 | 20.76 | **10.98** |
| | ResNet-50 | 96.20 | 19.52 | 16.36 | 24.59 | 19.76 | 20.33 | **9.53** |
| | VGG-11 | 95.50 | 21.26 | 30.08 | 31.16 | 19.98 | 20.62 | **11.57** |
| | DenseNet-121 | 96.22 | 17.34 | 22.33 | 38.68 | 18.68 | 30.31 | **10.80** |
| | ViT | 95.59 | 19.59 | 19.24 | 20.31 | 19.52 | 19.32 | **10.10** |

Table 2: Test accuracy (%) of UE methods across different backbones under the unfiltered setting.

| Dataset | Backbone | Clean | EMN | LSP | TUE | GUE | PUE | FUSE |
|---------|----------|-------|-----|-----|-----|-----|-----|------|
| CIFAR-10 | ResNet-18 | 94.66 | 16.42 | 13.54 | 10.03 | 13.25 | 10.62 | **9.41** |
| | ResNet-50 | 94.37 | 11.83 | 12.82 | 12.17 | 12.84 | 10.00 | **9.54** |
| | VGG-11 | 95.29 | 29.39 | 17.16 | 17.45 | 13.64 | 26.38 | **10.17** |
| | DenseNet-121 | 95.12 | 14.71 | 16.38 | 15.51 | 12.15 | 10.31 | **10.09** |
| | ViT | 72.34 | 29.68 | 16.23 | 15.29 | 12.75 | 19.43 | **10.07** |
| CIFAR-100 | ResNet-18 | 76.27 | 3.95 | 9.00 | 1.21 | 8.35 | 2.62 | **1.10** |
| | ResNet-50 | 70.48 | 3.80 | 13.63 | 9.56 | 4.89 | 2.03 | **1.46** |
| | VGG-11 | 67.89 | 7.13 | 20.16 | 6.90 | 6.09 | **2.02** | 2.49 |
| | DenseNet-121 | 74.51 | 4.75 | 12.70 | 12.82 | 3.56 | 3.01 | **1.19** |
| | ViT | 42.14 | 12.99 | 23.96 | 23.95 | 7.56 | 4.72 | **1.46** |
| SVHN | ResNet-18 | 96.05 | 9.05 | **7.35** | 9.12 | 13.70 | 14.21 | 8.11 |
| | ResNet-50 | 96.25 | 8.94 | 7.88 | 9.17 | 9.72 | 9.69 | **7.60** |
| | VGG-11 | 95.29 | 23.44 | 13.11 | 13.00 | 9.77 | 11.84 | **9.34** |
| | DenseNet-121 | 96.36 | **9.10** | 12.55 | 13.23 | 9.67 | 11.07 | 10.21 |
| | ViT | 95.81 | 10.32 | 14.06 | 9.59 | 8.83 | 7.76 | **6.55** |

thereby reinforcing its generality beyond the limitations of prior methods and ensuring consistent protection across diverse scenarios.

## 4.3 EVALUATION UNDER VARYING CUTOFF FREQUENCIES

To further assess effectiveness against spectral filtering, we evaluate different UE methods under low-pass filtering with varying cutoff frequencies. As shown in Figure 3a and Figure 3b, our proposed FUSE consistently reduces test accuracy to near random-guessing levels across all cutoff values on both CIFAR-10 and CIFAR-100. In particular, FUSE maintains test accuracy below 10% on CIFAR-100, demonstrating that the generated perturbations remain effective even under strong spectral suppression. In contrast, baseline methods show a marked decline in unlearnability when the cutoff is small, indicating that their perturbations primarily exploit high-frequency information. Notably, on CIFAR-100, TUE and PUE exhibit a non-monotonic trend where test accuracy initially increases before dropping at higher cutoffs. This occurs because their perturbations lack unlearnable effects in the low-frequency region, while the preserved semantic information of clean images becomes more

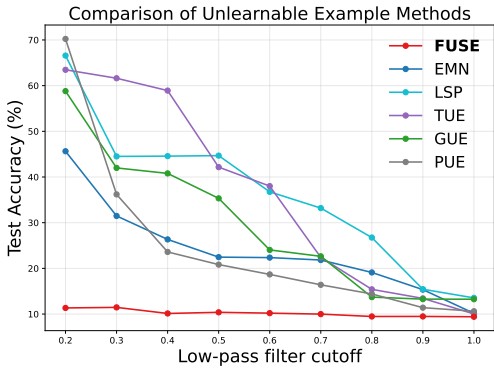
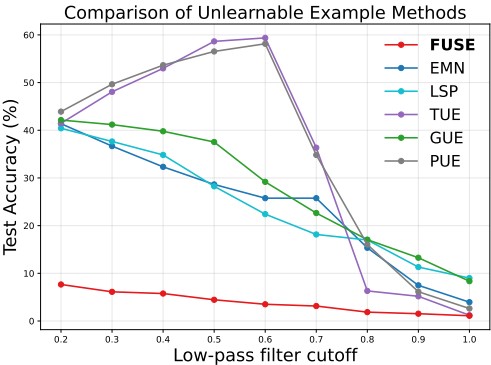

(a) Test accuracy (%) of UEs under low-pass filtering with varying cutoff frequencies on CIFAR-10.

(b) Test accuracy (%) of UEs under low-pass filtering with varying cutoff frequencies on CIFAR-100.

Figure 3: Test accuracy (%) ↓ of unlearnable example methods under low-pass filtering with varying cutoff frequencies on CIFAR-10 (a) and CIFAR-100 (b), using ResNet-18 as the backbone.

dominant as the cutoff increases. These results highlight that FUSE produces spectrum-agnostic perturbations, successfully overcoming the spectral fragility of prior approaches.

## 4.4 CROSS-ARCHITECTURE

We further evaluate the cross-architecture transferability of FUSE, which is essential for assessing whether UEs remain effective when applied to unseen target models. Using ResNet-18 as the surrogate model, we evaluate its ability to render other models unlearnable, including ResNet-50, VGG-11, DenseNet-121, and ViT. As shown in Table 3, FUSE consistently achieves the lowest test accuracy across all architectures, outperforming prior UE methods. These results demonstrate that FUSE is not overfitted to its surrogate model and preserves strong unlearnability even under architectural shifts.

Table 3: Test Cross-Architecture accuracy on CIFAR-10, where ResNet-18 is used as the surrogate model.

| Method | Network Architecture | | | |
|--------|-----------|--------|-------------|------|
|        | ResNet-50 | VGG-11 | DenseNet-121 | ViT |
| EMN    | 17.90     | 29.30  | 18.60       | 24.37 |
| TUE    | 13.41     | 26.29  | 17.39       | 26.29 |
| GUE    | 12.97     | 13.72  | 13.71       | 16.77 |
| PUE    | 12.57     | 27.71  | 14.04       | 20.07 |
| **FUSE** | **11.39** | **11.67** | **11.95** | **10.87** |

## 4.5 ABLATION STUDY

Table 4: Ablation studies on CIFAR-10, using ResNet-18 as the backbone. Acc@x denotes test accuracy under low-pass filtering with cutoff x.

| (a) Effect of different modules. | | | (b) Effect of frequency split radius $r_c$. | | | |
|--------|-------------|-------------|-------|-------------|-------------|-------------|
| Cutoff | Acc@0.5 ↓ | Acc@1.0 ↓ | $r_c$ | Acc@0.2 ↓ | Acc@0.4 ↓ | Acc@0.5 ↓ |
| w/o RSM | 57.35 | 11.13 | 0.2 | 11.72 | 11.06 | 10.93 |
| w/o entropy | 15.13 | 10.79 | 0.5 | **10.21** | **10.14** | **10.13** |
| w/o CBG | 17.00 | 11.50 | 0.8 | 15.53 | 15.36 | 14.70 |
| **Full FUSE** | **10.13** | **9.41** | | | | |

We then analyze the contribution of each module through ablation experiments (Table 4a). When Random Spectral Masking is removed, accuracy at cutoff $0.5$ increases sharply to $57.35\%$, showing that perturbations collapse into high frequencies and lose their unlearnability once low-pass filtering is applied. Removing the entropy regularizer also degrades performance, indicating that without explicitly spreading perturbation energy across the spectrum, perturbations become more localized and fragile. In contrast, removing Cross-Band Guidance leads to a more pronounced degradation under low-pass filtering, reflecting the importance of regulating low-frequency perturbations. Overall, the full FUSE consistently achieves the lowest accuracy, demonstrating that RSM, entropy regularization, and CBG are complementary: RSM enforces frequency diversity, entropy prevents spectral concentra-

tion, and CBG stabilizes cross-band interactions, together yielding spectrum-agnostic perturbations that remain effective under filtering.

We also analyze the choices of the frequency split radius $r_c$, as shown in Table 4b. A small value ($r_c = 0.2$) provides insufficient low-frequency semantics, leading to unstable guidance signals and weaker unlearnability. In contrast, a large value ($r_c = 0.8$) overly reduces the distinction between low- and high-frequency bands, weakening cross-band interaction. The best performance is consistently achieved at a moderate split ($r_c = 0.5$), where the separation is sufficient to enforce interaction while still maintaining semantic stability.

To assess the effect of the structural consistency loss, we vary its weight $\lambda$ in the CBG objective on the CIFAR-10 dataset (Figure 4). When $\lambda$ is very small, the consistency loss is too weak, leaving high-frequency perturbations insufficiently constrained and leading to unstable low-frequency guidance. As a result, performance under unfiltered setting training is largely unaffected, but unlearnability degrades under low-pass filtering. As $\lambda$ increases, the structural loss regularizes high-frequency perturbations and reinforces low-frequency effects, improving unlearnability in both settings. Moderate values yield the strongest results by balancing semantic

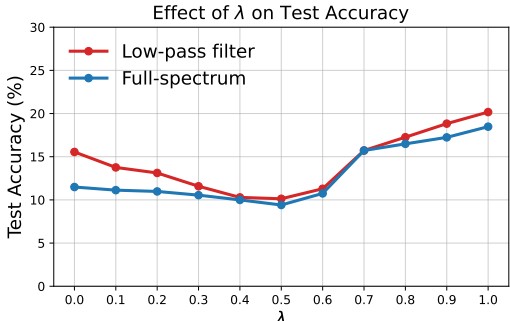

Figure 4: Effect of the weight $\lambda$ on test accuracy.

guidance and structural alignment, while large $\lambda$ makes the structural loss dominate, weakening the guidance from high- to low-frequency perturbations and thus reducing overall unlearnability.

### 4.6 Resistance to Defense Strategies

To further evaluate robustness, we test the performance of different UE methods against common defense strategies, including data augmentation (DeVries & Taylor, 2017; Yun et al., 2019; Zhang et al., 2018) and Adversarial Training (Madry, 2018). As shown in Table 5, while some baselines retain partial effectiveness under certain defenses, their unlearnability generally collapses once different types of defenses are applied. In contrast, FUSE consistently main-

Table 5: Test accuracy of ResNet-18 in CIFAR-10 dataset, where ResNet-18 is also used as the backbone. "AT" denotes Adversarial Training.

| Method | w/o | Cutout | CutMix | Mixup | AT |
|---|---|---|---|---|---|
| Clean | 94.66 | 95.10 | 95.50 | 95.01 | 84.99 |
| EMN | 10.16 | 20.63 | 26.19 | 32.83 | 84.80 |
| LSP | 13.54 | 19.87 | 20.89 | 26.99 | 84.59 |
| AR | 11.75 | 12.36 | 18.02 | 14.59 | 83.17 |
| OPS | 15.56 | 61.68 | 76.40 | 33.13 | 11.08 |
| **FUSE** | **9.41** | **10.13** | **13.09** | **14.35** | **10.80** |

tains low test accuracy across all settings, often close to random-guessing levels, demonstrating that its spectrum-agnostic design prevents models from circumventing perturbations through either augmentation or adversarial retraining. Additional results against JPEG are shown in Section D.5.

## 5 Conclusion

In this paper, we introduced Full-spectrum Unlearnable Examples via Spectral Equalization (FUSE), a spectrum-agnostic framework for protecting training data against unauthorized learning. Unlike existing methods whose effectiveness critically depends on high-frequency components and thus collapses under low-pass filtering, FUSE explicitly distributes perturbations across the entire frequency range and enforces cross-band consistency. Through the integration of Random Spectral Masking (RSM) and Cross-Band Guidance (CBG), FUSE generates perturbations that remain unlearnable under diverse filtering operations while preserving semantic fidelity. Extensive experiments across multiple datasets and architectures demonstrate that FUSE achieves consistently stronger and more stable unlearnability than prior approaches. Our results highlight the importance of frequency-domain modeling for data protection and suggest promising directions for future work, such as designing privacy-preserving mechanisms that remain effective under distribution shifts and adaptive defenses.

## ETHICAL STATEMENT

Our work introduces Full-spectrum Unlearnable Examples via Spectral Equalization (FUSE), a framework designed to protect data privacy by generating perturbations that prevent unauthorized models from learning useful semantics. While our goal is to strengthen privacy protection, we recognize that unlearnable perturbations could, in principle, be misused in adversarial contexts to interfere with model training. We emphasize that FUSE should be applied solely for ethical data protection purposes, and any use must comply with relevant legal frameworks, community standards, and dataset licenses. All datasets used in this study (CIFAR-10, CIFAR-100, SVHN, and ImageNet) are publicly available benchmarks, and our usage complies with their respective licenses and intended research purposes.

## REPRODUCIBILITY STATEMENT

To ensure reproducibility, we provide complete implementation details of FUSE, including the perturbation generator, training settings, and evaluation protocols across datasets. All hyperparameters (e.g., cutoff radius, perturbation bounds, and balancing coefficients) are explicitly specified in the paper or appendix. Code for training, evaluation, and ablation studies will be made available in the supplementary materials, along with detailed documentation on environment configuration and instructions for reproducing all reported results.

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

# APPENDIX

## A    USE OF LARGE LANGUAGE MODELS (LLMS)

We declare that Large Language Models (LLMs) were used exclusively as auxiliary tools to aid the writing of this paper, primarily for polishing, grammar checking, and improving readability. LLMs were not involved in conceptual design, theoretical formulation, experimental implementation, or result analysis. All research contributions, including methodology, experiments, and conclusions, were conceived, conducted, and validated entirely by the authors.

## B    THE ARCHITECTURE OF PERTURBATION GENERATOR

In the main paper, we devise a generator to produce transferable perturbations and craft unlearnable examples. We denote our perturbation generator as $\mathcal{G}$, which employs a standard encoder-decoder architecture. This structure comprises three down-sampling convolution layers, four residual blocks (He et al., 2016), and three transposed convolution layers. The detailed architecture is shown in Table 6.

Table 6: The architecture of the perturbation generator.

| Block Name | Layer | Number |
|---|---|---|
| *Down-sampling layers* | | |
| Conv | Conv ($3 \times 3$) InstanceNorm ReLU | $\times 3$ |
| *Bottleneck layers* | | |
| Residual | ReflectionPad Conv ($3 \times 3$) BatchNorm ReLU ReflectionPad Conv ($3 \times 3$) BatchNorm | $\times 4$ |
| *Up-sampling layers* | | |
| ConvTranspose | ConvTranspose ($3 \times 3$) InstanceNorm ReLU | $\times 2$ |
| ConvTranspose | ConvTranspose ($6 \times 6$) Tanh | $\times 1$ |

## C    MORE IMPLEMENTATION DETAILS

**Implementation of FUSE.** For the perturbation generator, we adopt the Adam optimizer (Kingma & Ba, 2015) with an initial learning rate of 0.001. For surrogate models, we use the SGD optimizer (Le-Cun et al., 1998) with an initial learning rate of 0.1, applied to ResNet-18, ResNet-50, VGG-11, DenseNet-121, and ViT. To evaluate unlearnability, we employ the cross-entropy loss as the training objective in unauthorized supervised learning. Our method is a class-wise UE method and we add perturbations to the entire training dataset, which is the common practice in UE literature, to make all user data unexploitable by unauthorized training.

**Implementation of Baselines.** For a fair comparison, we adopt representative unlearnable example (UE) methods as baselines. Specifically, EMN (Huang et al., 2021), LSP (Yu et al., 2022), and PUE (Wang et al., 2024) follow a *class-wise* design, where perturbations are generated from class-level aggregation so that samples within the same class share similar perturbation patterns. In contrast, TUE (Ren et al., 2022) and GUE (Liu et al., 2024a) employ a *sample-wise* approach, where each

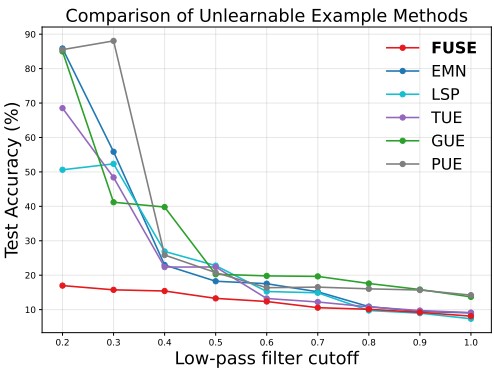 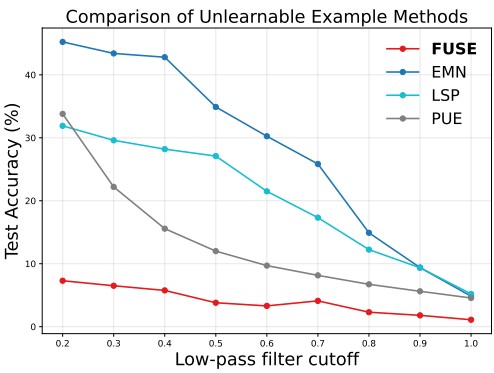

(a) Test accuracy (%) of UEs under low-pass filtering with varying cutoff frequencies on SVHN.

(b) Test accuracy (%) of UEs under low-pass filtering with varying cutoff frequencies on ImageNet$^*$.

Figure 5: Test accuracy (%) ↓ of unlearnable example methods under low-pass filtering with varying cutoff frequencies on SVHN (a) and ImageNet$^*$ (b), using ResNet-18 as the backbone.

image is perturbed independently. We reproduce all baselines using their official implementations or configurations reported in the original papers, and tune the perturbation budget to align with our setting ($\epsilon = 8/255$) for consistency.

# D  ADDITIONAL EXPERIMENTS

## D.1  ADDITIONAL RESULTS ON SVHN AND IMAGENET

To further evaluate effectiveness against spectral filtering, we extend our analysis to SVHN and ImageNet$^*$ (the first 100 classes of ImageNet (Russakovsky et al., 2015)) under low-pass filtering with varying cutoff frequencies. As shown in Figure 5a and Figure 5b, FUSE consistently achieves near-random test accuracy across a wide range of cutoffs, indicating that its perturbations remain effective even under strong spectral suppression.

On SVHN (Figure 5a), when the cutoff is as small as 0.2, most baselines still yield relatively high accuracy (above $60\%$), whereas FUSE immediately reduces performance to around $10\%$, close to random guessing. As the cutoff increases, the accuracies of baseline methods rapidly decline and eventually converge to FUSE's level. At higher cutoffs (e.g., $> 0.7$), a few methods slightly outperform FUSE, but all methods remain in the vicinity of random-guessing accuracy, rendering the differences negligible.

On ImageNet$^*$ (Figure 5b), the contrast is even more pronounced. FUSE consistently suppresses accuracy below $10\%$ across nearly all cutoffs, while baselines such as EMN and LSP remain significantly higher (30–$50\%$) under small cutoffs (0.2–0.4), clearly demonstrating their dependence on high-frequency perturbations. As the cutoff increases, their accuracy declines but still remains above FUSE. This pattern confirms that FUSE perturbations are not confined to a particular spectral band and remain effective under both strong and weak filtering.

Overall, these results highlight the spectrum-agnostic property of FUSE. By maintaining unlearnability across datasets of different scale and complexity (SVHN and ImageNet$^*$), FUSE preserves its unlearnable effect under low-pass filtering, whereas prior approaches degrade substantially when high-frequency perturbations are suppressed.

## D.2  ABLATION STUDIES ON PERTURBATION STRENGTH

We further investigate the role of perturbation strength by varying the bound $\epsilon$ in Eq. 14 on CIFAR-10, CIFAR-100, and SVHN under both unfiltered and low-pass filtering conditions (cutoff = 0.5). As shown in Table 7, increasing $\epsilon$ consistently improves unlearnability within each dataset, as reflected by the monotonic reduction in test accuracy toward the random-guess level (10% for CIFAR-10 and SVHN, 1% for CIFAR-100).

Table 7: Effect of perturbation strength $\epsilon$ with and without low-pass filtering (cutoff $= 0.5$). "LPF" means low-pass filtering.

| $\epsilon$ | No LPF (Unfiltered) | | | LPF @ 0.5 | | |
|---|---|---|---|---|---|---|
| | CIFAR-10 | CIFAR-100 | SVHN | CIFAR-10 | CIFAR-100 | SVHN |
| 2/255 | 11.00 | 1.59 | 11.31 | 15.53 | 5.77 | 14.61 |
| 4/255 | 10.43 | 1.27 | 10.33 | 13.82 | 4.83 | 11.11 |
| 8/255 | 9.41 | 1.10 | 8.11 | 10.13 | 4.45 | 10.98 |
| 16/255 | 9.36 | 0.99 | 7.98 | 10.08 | 4.14 | 9.30 |
| 32/255 | 8.53 | 0.71 | 5.96 | 9.12 | 2.30 | 7.51 |

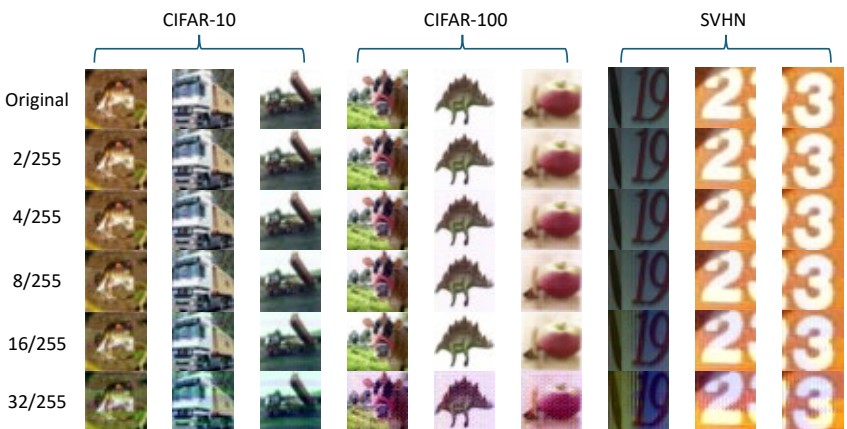

Figure 6: Visual examples of FUSE-generated unlearnable images under different perturbation strengths.

On CIFAR-10, accuracy decreases from $11.0\%$ at $\epsilon = 2/255$ to $9.4\%$ in the unfiltered setting, and from $15.5\%$ to $10.1\%$ under low-pass filtering. On CIFAR-100, accuracy drops from $1.6\%$ to below $1.0\%$ as $\epsilon$ increases, which is already close to random guessing, showing that even mild perturbations suffice to induce unlearnability. A similar trend is observed on SVHN, where accuracy decreases from $11.3\%$ at $\epsilon = 2/255$ to $8.0\%$ at $\epsilon = 16/255$ in the unfiltered setting, and from $14.6\%$ to $9.3\%$ with filtering.

These results confirm that stronger perturbations remain effective even when high-frequency components are suppressed. Nevertheless, larger $\epsilon$ inevitably reduces visual imperceptibility, highlighting the inherent trade-off between perturbation strength and image quality. Importantly, the consistent patterns across datasets and filtering conditions indicate that FUSE's unlearnability is not tied to a specific perturbation budget or frequency band.

To complement the quantitative results in Table 7, we further provide visualizations of FUSE-generated unlearnable examples under different perturbation strengths. As shown in Figure 6, small perturbations remain visually indistinguishable but provide weaker unlearnable effects. Larger perturbations produce stronger suppression of learnability but also lead to noticeable visual degradation.

These visual trends align well with the quantitative results: increasing $\epsilon$ consistently lowers the test accuracy of the defender toward random-guess levels, while the perceptual distortion gradually increases. Together, the visual and numerical evidence illustrate the intrinsic trade-off between perturbation intensity and imperceptibility.

### D.3 CROSS-DATASET TRANSFERABILITY OF UEs

We further evaluate cross-dataset unlearnability by generating perturbations on one dataset (CIFAR-10, CIFAR-100, or SVHN) and applying them to other datasets. Results are reported for both the

Table 8: Cross-dataset transferability in the unfiltered case. Perturbations are generated on the source dataset and applied to the target datasets, where ResNet-18 is used as the surrogate model.

| Source | Target | EMN | LSP | TUE | GUE | PUE | FUSE |
|---|---|---|---|---|---|---|---|
| CIFAR-10 | CIFAR-100 | 21.80 | 9.35 | 5.10 | 3.87 | 8.46 | **2.18** |
| | SVHN | 24.72 | **7.77** | 12.93 | 8.17 | 12.01 | 9.94 |
| CIFAR-100 | CIFAR-10 | 27.27 | 24.16 | 94.31 | 94.28 | 11.61 | **9.98** |
| | SVHN | 9.64 | 17.03 | 96.02 | 95.87 | 18.58 | **8.39** |
| SVHN | CIFAR-10 | 14.31 | 38.50 | 93.91 | 94.31 | 10.23 | **10.04** |
| | CIFAR-100 | 6.25 | 38.51 | 69.42 | 48.37 | 6.04 | **5.25** |

Table 9: Cross-dataset transferability under low-pass filter. Perturbations are generated on the source dataset and applied to the target datasets, where ResNet-18 is used as the surrogate model.

| Source | Target | EMN | LSP | TUE | GUE | PUE | FUSE |
|---|---|---|---|---|---|---|---|
| CIFAR-10 | CIFAR-100 | 27.16 | 19.75 | 62.96 | 59.57 | 20.05 | **5.89** |
| | SVHN | 25.40 | 25.29 | 95.87 | 95.74 | 15.73 | **11.09** |
| CIFAR-100 | CIFAR-10 | 32.15 | 33.81 | 90.01 | 88.44 | 67.75 | **17.08** |
| | SVHN | 19.72 | 20.70 | 95.93 | 95.30 | 26.30 | **9.75** |
| SVHN | CIFAR-10 | 63.67 | 42.36 | 89.98 | 89.17 | 42.62 | **13.12** |
| | CIFAR-100 | 62.61 | 39.92 | 61.38 | 60.14 | 55.70 | **16.29** |

unfiltered setting (Table 8) and under low-pass filtering at test time (Table 9). Since LSP (Yu et al., 2022) and TUE (Ren et al., 2022) generate fixed perturbations, we first resample these perturbations before applying them to the images to ensure a fair comparison. When the number of samples differs, we generate new perturbation samples using a uniform sampling strategy (Ren et al., 2022) or randomly cropping the saved perturbations to remove redundant portions, thereby ensuring compatibility with the target dataset.

Across both conditions, FUSE generally achieves the lowest transfer accuracy, indicating the strongest unlearnable effect. In particular, when perturbations are generated on CIFAR-100 and transferred to CIFAR-10, FUSE reduces accuracy to 9.98% in the unfiltered case, while baselines such as TUE and GUE exceed 90%. Even after low-pass filtering, FUSE maintains low transfer accuracy (17.08%), whereas competing methods remain above 60%. A similar advantage is observed in the SVHN-to-CIFAR-100 transfer, where FUSE suppresses accuracy to 5.25% in the unfiltered setting and 16.29% with low-pass filtering, again far below other approaches.

Other transfer pairs show the same trend. For example, from CIFAR-10 to SVHN, FUSE achieves 9.94% (unfiltered) and 11.09% (low-pass), while the baselines range from 8–25% in the former but surge above 90% in the latter, exposing their vulnerability to spectral filtering. Similarly, when transferring from SVHN to CIFAR-10, FUSE maintains accuracy near 10% under both conditions, whereas TUE and GUE exceed 90%.

Overall, these results demonstrate that FUSE-generated perturbations transfer more reliably across datasets and remain effective even under spectral suppression. This advantage arises because natural images share highly similar frequency distributions (Field, 1987; Ruderman, 1994), and our design ensures effectiveness across the entire spectrum. Consequently, when facing cross-dataset transfer settings, FUSE maintains strong unlearnability, whereas prior methods exhibit limited transferability in the unfiltered case and collapse once low-pass filtering is applied.

Table 10: Test accuracy (%) for different unlearnable percentages $p$ under a low-pass filtering operation.

| Percentages $p$ | 10% | 20% | 30% | 40% | 50% | 60% | 70% | 80% | 90% | 100% |
|---|---|---|---|---|---|---|---|---|---|---|
| Clean $(1-p)$ | 85.39 | 85.26 | 84.23 | 83.70 | 83.01 | 82.24 | 81.13 | 77.30 | 75.49 | – |
| EMN | 90.53 | 87.55 | 87.13 | 86.50 | 85.72 | 85.22 | 84.95 | 83.93 | 83.28 | 23.56 |
| LSP | 90.30 | 88.90 | 88.52 | 87.93 | 87.81 | 87.05 | 86.58 | 85.96 | 85.32 | 54.16 |
| TUE | 90.04 | 87.88 | 88.35 | 87.82 | 87.59 | 87.58 | 87.43 | 86.79 | 85.03 | 52.16 |
| GUE | 90.01 | 89.24 | 89.04 | 88.71 | 88.14 | 87.62 | 87.28 | 86.72 | 85.78 | 41.97 |
| PUE | 89.51 | 88.97 | 88.54 | 88.22 | 87.88 | 87.40 | 87.26 | 86.72 | 84.04 | 23.60 |
| **FUSE** | **87.29** | **86.41** | **85.92** | **85.06** | **84.94** | **84.13** | **83.12** | **79.38** | **77.76** | **10.13** |

Table 11: Test accuracy (%) for different unlearnable percentages $p$ without any filtering applied.

| Percentages $p$ | 10% | 20% | 30% | 40% | 50% | 60% | 70% | 80% | 90% | 100% |
|---|---|---|---|---|---|---|---|---|---|---|
| Clean $(1-p)$ | 94.01 | 93.75 | 93.10 | 92.56 | 92.42 | 89.77 | 86.44 | 84.30 | 82.64 | – |
| EMN | 95.29 | 94.24 | 93.88 | 92.99 | 92.87 | 91.10 | 88.52 | 87.23 | 85.21 | 16.42 |
| LSP | 95.70 | **94.04** | 93.79 | 93.10 | 92.91 | 91.96 | 88.93 | 86.29 | 85.73 | 13.54 |
| TUE | 95.18 | 94.84 | 94.35 | 93.88 | 92.68 | 91.05 | 88.19 | 86.62 | 85.39 | 10.03 |
| GUE | 95.78 | 95.07 | 94.48 | 93.94 | 93.15 | 90.41 | 85.40 | **83.99** | 83.12 | 13.25 |
| PUE | 95.10 | 94.84 | 94.30 | 93.69 | 93.24 | 91.74 | 88.75 | 86.10 | 85.07 | 10.62 |
| **FUSE** | **94.79** | 94.21 | **93.64** | **92.37** | **92.28** | **90.11** | **85.38** | 84.66 | **82.93** | **9.41** |

## D.4 MORE ANALYSES OF DIFFERENT UNLEARNABLE PERCENTAGES.

In addition to the main experiments, we further investigate how the proportion of perturbed training samples affects unlearnability. While the standard UE setting perturbs the entire training set, practical applications may involve scenarios where only a given percentage of the training data ($p$) is protected. To this end, we follow the different unlearnable percentages protocol and vary the perturbed-data ratio (Huang et al., 2021; Yu et al., 2022). Specifically, for each unlearnable percentage, we train two models. The first model uses both the clean subset and the unlearnable subset as its training data, and the second one only uses the clean subset. The difference between the performances of those two models represents how much semantic information the first model gains from the UEs. A small performance gap indicates the first model gains little information from the UEs. As shown in Table 10 (with a cutoff radius of 0.5) and Table 11 (unfiltered), existing UE methods reduce protection when only part of the data is perturbed. In contrast, FUSE consistently achieves accuracy closest to the clean baseline across most poisoning ratios, indicating that the model effectively ignores perturbed samples even when the unlearnable portion is limited.

## D.5 MORE ANALYSES OF JPEG COMPRESSION DEFENSES.

We further evaluate the robustness of FUSE under different JPEG compression levels, combined with a low-pass filter. JPEG compression removes high-frequency components and introduces frequency artifacts, making it a challenging perturbation scenario for unlearnable examples. Since FUSE proposes full-spectrum unlearable perturbations, rather than restricting perturbations to a single frequency band, FUSE is robust to the JPEG compression compared with other UE methods. As shown in Table 12 and Table 13, FUSE consistently maintains near random-guess accuracy across a wide range of JPEG quality levels, demonstrating that our method remains effective even under strong frequency-domain distortions. To verify this, we further demonstrate the impact of JPEG compression (Q=70) and low-pass filtering using t-SNE visualizations in Figure 7. The results show that under full-spectrum conditions, FUSE's perturbations result in a low accuracy, lower than other methods. When low-pass filtering is applied after JPEG compression, FUSE's performance remains close to random-guess level. These results further support our claim that FUSE is more robust to JPEG compression than other methods.

Table 12: Test accuracy (%) on CIFAR-10 under different JPEG compression qualities.

| JPEG Compression | 10 | 20 | 30 | 40 | 50 | 60 | 70 | 80 | 90 |
|---|---|---|---|---|---|---|---|---|---|
| Clean | 82.24 | 84.40 | 85.05 | 85.84 | 86.09 | 86.60 | 87.25 | 87.67 | 87.90 |
| EMN | 81.93 | 80.30 | 76.37 | 69.07 | 69.07 | 65.89 | 62.01 | 42.56 | 25.48 |
| LSP | 82.03 | 83.14 | 81.59 | 78.89 | 76.57 | 72.27 | 68.17 | 62.72 | 57.48 |
| TUE | 59.53 | 58.63 | 54.10 | 57.13 | 56.25 | 57.15 | 55.89 | 54.46 | 54.62 |
| GUE | 78.88 | 78.24 | 74.53 | 72.76 | 71.26 | 66.32 | 60.59 | 50.41 | 43.38 |
| PUE | 54.84 | 45.99 | 43.78 | 37.63 | 31.73 | 29.80 | 27.40 | 25.89 | 23.76 |
| **FUSE** | **54.56** | **35.16** | **31.60** | **27.32** | **25.41** | **20.77** | **18.90** | **12.87** | **10.98** |

Table 13: Test accuracy (%) on ImageNet* under different JPEG compression qualities.

| JPEG Compression | 10 | 20 | 30 | 40 | 50 | 60 | 70 | 80 | 90 |
|---|---|---|---|---|---|---|---|---|---|
| Clean | 54.60 | 58.30 | 58.40 | 59.30 | 59.40 | 59.80 | 60.00 | 60.60 | 62.30 |
| **FUSE** | **5.70** | **9.50** | **3.30** | **2.80** | **6.70** | **8.60** | **7.70** | **5.50** | **6.00** |

## D.6 ABLATION ON FREQUENCY SPLIT RADIUS

The frequency split radius $r_c$ controls the proportion of low-frequency components retained in the spectral decomposition of FUSE. To verify whether the optimal $r_c$ is dataset-dependent, we conduct a comprehensive ablation study on four datasets of different resolutions and characteristics: CIFAR-10, CIFAR-100, SVHN, and ImageNet*. As shown in Table 14, the optimal $r_c$ consistently lies in a narrow range around 0.5, and the performance variation within this range is small. This indicates that the choice of $r_c$ is stable across datasets and not dataset-specific.

Furthermore, we applied Bayesian Optimization to automatically search for the optimal $r_c$ on each dataset (Table 15). The automatically identified values also converge near 0.5, and the resulting accuracies differ from the fixed setting $r_c = 0.5$ by less than 0.1%–0.2%. These results confirm that the manually chosen $r_c = 0.5$ is close to the global optimum and not the result of manual tuning. While selecting the dataset-wise optimal $r_c$ yields marginal improvements, the overall performance differences are small, demonstrating that FUSE is not sensitive to this hyperparameter. For simplicity and reproducibility, we therefore adopt $r_c = 0.5$ as a robust default choice that generalizes well across datasets.

Table 14: Ablation on the frequency split radius $r_c$ across four datasets.

| $r_c$ | CIFAR-10 | CIFAR-100 | SVHN | ImageNet* |
|---|---|---|---|---|
| 0.2 | 10.93 | 5.03 | 11.15 | 4.52 |
| 0.3 | 10.84 | 4.97 | 11.01 | 4.16 |
| 0.4 | 10.32 | 4.78 | **10.17** | 3.56 |
| 0.5 | **10.13** | **4.45** | 10.98 | **3.20** |
| 0.6 | 10.38 | 4.46 | 11.37 | 3.94 |
| 0.7 | 11.43 | 5.65 | 16.57 | 5.15 |
| 0.8 | 14.70 | 6.24 | 18.57 | 6.24 |

## D.7 EFFECT OF THE NUMBER OF FREQUENCY BANDS

To examine whether a simple high-/low-frequency split may be insufficient, we conducted an additional ablation study on ImageNet* where the frequency domain is divided into three bands (low, mid, high). We further applied our CBG module (Section 3.3) to all pairwise band interactions (low-to-mid, low-to-high, and mid-to-high). The results are summarized in Table 16.

It can be observed that, while modest, three-band split indeed leads to stronger unlearnability. On the other hand, we note that both two-band and three-band designs achieve accuracies that are close

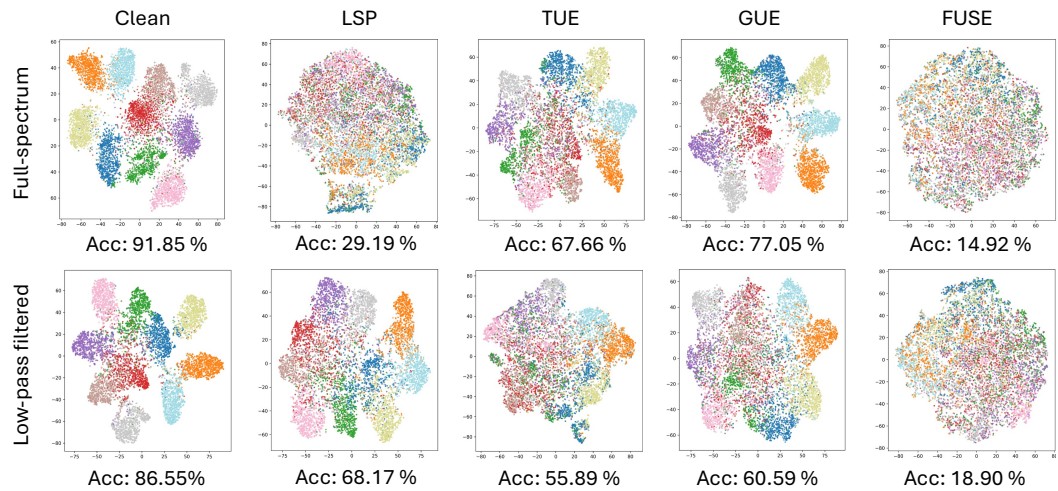

Figure 7: t-SNE visualization under **JPEG compression** (Q=70) with and without low-pass filtering.

Table 15: Optimal $r_c$ identified by Bayesian Optimization and the corresponding unlearnable accuracy.

|  | CIFAR-10 | CIFAR-100 | SVHN | ImageNet* |
|---|---|---|---|---|
| BO-selected $r_c$ | 0.47 | 0.51 | 0.42 | 0.46 |
| Test Accuracy | 10.06 | 4.36 | 10.01 | 3.12 |

to random guess, whereas latter incurs higher computational cost, indicating a trade-off between effectiveness and efficiency.

Table 16: Comparison between two-band split and three-band split on ImageNet*.

| Method | Low-pass filter | Unfiltered | Training time (s/epoch) |
|---|---|---|---|
| Two-band split | 3.20 | 1.12 | **21.24** |
| Three-band split | **2.84** | **1.02** | 32.03 |

## D.8  HIGH-PASS FILTERING ANALYSIS

We further evaluate FUSE under high-pass filtering, which removes low-frequency components and retains only high-frequency details. As shown in Table 17, FUSE consistently maintains near random-guess accuracy across all cutoff values, demonstrating robustness to both low-frequency (Table 1) and high-frequency removal and supporting our spectrum-agnostic claim. Notably, the observation that prior UE methods also remain effective under high-pass filtering is consistent with our analysis in the paper. In contrast, their failure under low-pass filtering (as shown in Table 1, Figure 3a and Figure 3b) arises because the high-frequency components are suppressed.

## D.9  MORE ABLATION ON STRUCTURAL SIMILARITY.

The structure guidance in Section 3.3 includes cosine similarity and structural similarity. To isolate the contribution of structural similarity, we conduct an ablation study on unfiltered CIFAR-10 Dataset by removing the structural similarity term while keeping all other components unchanged. As shown in Table 18, removing structural similarity leads to degraded the quality of UEs, reflected by lower PSNR/SSIM and higher LPIPS/MSE. These results confirm that the structural similarity term enhances visual fidelity and structural preservation, complementing the cosine-similarity term.

Table 17: Unlearnability performance under high-pass filtering with different cutoff.

| High-frequency cutoff | 0.1 | 0.2 | 0.3 | 0.4 | 0.5 | 0.6 | 0.7 | 0.8 | 0.9 |
|---|---|---|---|---|---|---|---|---|---|
| Clean | 94.69 | 94.43 | 93.65 | 93.45 | 92.30 | 91.56 | 91.01 | 90.54 | 90.41 |
| EMN | 13.71 | 11.98 | 11.14 | 10.41 | 10.46 | 10.04 | 10.91 | 11.03 | 14.12 |
| LSP | 13.47 | 12.04 | 11.81 | 10.85 | 9.81 | 11.53 | 15.55 | 19.54 | 23.93 |
| TUE | 10.24 | 10.51 | 10.74 | 10.55 | **9.55** | 10.25 | 11.23 | 13.41 | 15.38 |
| GUE | 13.40 | 13.76 | 12.73 | 16.01 | 10.47 | 10.03 | 10.22 | 10.13 | 12.73 |
| PUE | 10.92 | 10.46 | 11.07 | 10.45 | 10.03 | **9.66** | 11.49 | 13.57 | 11.92 |
| **FUSE** | **9.41** | **10.34** | **10.68** | **10.24** | 10.37 | 10.00 | **9.91** | **10.05** | **11.39** |

Table 18: Ablation study on the structural similarity component of CBG (CIFAR-10).

| Method | PSNR ↑ | SSIM ↑ | LPIPS ↓ | MSE ↓ |
|---|---|---|---|---|
| w/o structure | 37.17 | 0.9707 | 0.0374 | 0.000358 |
| FUSE | **39.35** | **0.9923** | **0.0163** | **0.000166** |

## D.10   MORE ANALYSIS OF TRAINING TIME AND COMPUTATIONAL RESOURCE REQUIREMENTS

Most existing UE methods directly store and add perturbations without any learnable generator. Therefore, for a fair comparison, we evaluate computational efficiency against the generator-based baseline GUE. Specifically, we feed each generator with a total of 50000 images from CIFAR-10 using a batch size of 512 to measure training time, inference time, GFLOPs, and parameters. All experiments are conducted on a consistent hardware setup comprising a NVIDIA L40S GPU. These results show that FUSE contains fewer parameters and achieves lower inference time and computational cost, highlighting its efficiency and practicality for real-world deployment.

Table 19: Comparison of computational efficiency between GUE and FUSE. FUSE achieves significantly lower training time, inference time, GFLOPs, and parameters.

| Method | Training time (s/epoch) | Inference time (ms/img) | GFLOPs | Parameters (M) |
|---|---|---|---|---|
| GUE | 321.10 | 0.194 | 3502.07 | 7.79 |
| **FUSE** | **21.24** | **0.004** | **1146.28** | **0.09** |

## E   VISUALIZATION

### E.1   TEST ACCURACY CURVES

In this section, we evaluate the test accuracy curves of unlearnable examples under low-pass filtering, using ResNet-18 (He et al., 2016) models trained on the perturbed CIFAR-10 dataset. As shown in Figure 8, most existing UE methods (e.g., EMN (Huang et al., 2021), GUE (Liu et al., 2024a), and PUE (Wang et al., 2024)) still retain non-trivial accuracy after filtering, and some even exhibit early accuracy peaks, which may lead to semantic leakage. In contrast, our proposed FUSE consistently maintains accuracy close to the random-guessing level of 10% from the first epoch and remains stable throughout training. These results suggest that FUSE achieves stronger and more persistent unlearnability, even under spectrum-suppressed conditions introduced by low-pass filtering.

### E.2   VISUALIZATION OF PERTURBATIONS UNDER LOW-PASS FILTERING

To further illustrate the effect of low-pass filtering on UEs, we visualize perturbed images from CIFAR-10, CIFAR-100, and SVHN under varying cutoffs (Figure 9). Notably, when the cutoff is 0.1, images become nearly structureless and contain little semantic content. For other cutoff values, semantic structures such as object shapes and digit outlines remain recognizable, indicating that low-

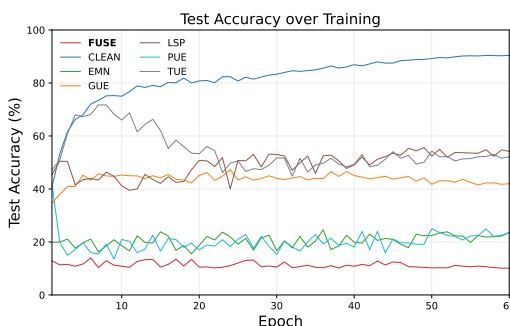

Figure 8: Test accuracy curves of ResNet-18 trained on perturbed CIFAR-10 under low-pass filtered UEs with different unlearnable example methods.

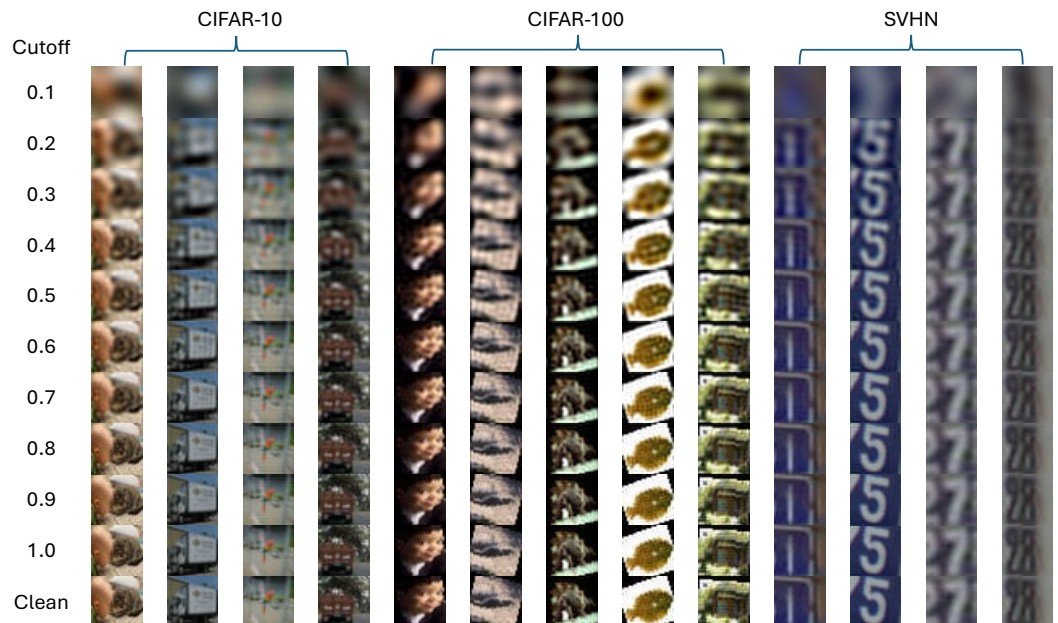

Figure 9: Visualization of perturbed images under different low-pass cutoffs (0.1–1.0) on CIFAR-10, CIFAR-100, and SVHN. Despite progressively removing high-frequency components, images retain recognizable semantic structures, which explains why models can still extract useful information from low-pass filtered data. Our perturbations remain visually similar to clean samples across all cutoffs, indicating strong imperceptibility. Notably, when the cutoff is 0.1, images become nearly structureless and contain little semantic content.

frequency information alone can still support partial learning. At the same time, our perturbations are visually indistinguishable from their clean counterparts, demonstrating that FUSE preserves imperceptibility while distributing perturbation effects across the spectrum.

