# OpenReview forum: "FUSE: Full‑spectrum Unlearnable Examples via Spectral Equalization"
_ICLR.cc/2026/Conference — Submitted to ICLR 2026_

### Official Review · Reviewer_Vrst · 2025-10-25

**Soundness:** 3
**Presentation:** 3
**Contribution:** 2
**Rating:** 6
**Confidence:** 3

**Summary:**

In this paper, to address the vulnerability of unlearnable sample generation methods,  authors propose Full Spectrum Unlearnable examples via Spectral Equalization (FUSE), which aims to generate spectrum-agnostic perturbations by equalizing the contributions from different bands and enforcing cross-band consistency.  Specifically, FUSE adopts a Random Spectral Masking (RSM) strategy during generator training, which randomly removes a contiguous frequency band, forcing the remaining bands to maintain unlearnability. In addition, FUSE further integrates Cross-Band Guidance (CBG), which enforces mutual consistency between high- and low-frequency components, thereby further enhancing low-frequency unlearnability and regulating high-frequency perturbations to preserve the semantic fidelity of images. Extensive experiments across multiple datasets, architectures, and spectral filtering demonstrate the strong protection achieved by FUSE.

**Strengths:**

1. The work verifies that existing unlearnable examples (UEs) generated by existing methods are vulnerable for low-pass filter. That indicates that  UEs rely on the high frequency components to achieve the unlearnable target.  Once these components are suppressed, deep models still can learn residual information of removed high frequency components. Through experiments on several datasets with different deep model, the proposed overcome the spectral vulnerability of existing methods.

2. This work propose a novel UEs generation method FUSE.FUSE is a novel frequency-agnostic framework designed to distribute perturbation effects across the entire frequency range through spectral equalization, ensuring that unlearnability is preserved even when specific frequency bands are suppressed.

3. The paper conducted extensive experiments on multiple datasets (CIFAR-10, CIFAR-100, SVHN), various model architectures (ResNet-18, ResNet-50, VGG-11, DenseNet-121, ViT), and different filtering scenarios. The experimental results show that FUSE consistently outperforms existing methods, achieving stronger non learnability in both low-pass filtering and unfiltered settings, while maintaining the imperceptibility of perturbations. For example, on the CIFAR-10 dataset, FUSE reduced the testing accuracy to 10.13% under low-pass filtering, which is much lower than other baseline methods under unfiltered settings. FUSE also achieved the lowest testing accuracy on most dataset architecture pairs, proving that its perturbation is still effective without spectral distortion. The paper also conducted ablation research to verify the contributions of RSM and CBG to FUSE performance, and analyzed the influence of hyperparameters such as frequency segmentation radius.

4. FUSE also evaluated common defense strategies such as data augmentation (Cutout, CutMix, Mixup) and adversarial training. The experimental results show that FUSE maintains a low testing accuracy under all settings, approaching the level of random guessing. This indicates that its spectrum independent design can effectively prevent the model from avoiding disturbances through reinforcement or adversarial retraining, demonstrating comprehensive resistance to a wide range of defense mechanisms.

5. The perturbations generated by FUSE exhibit stronger reliability when migrating across datasets, and remain effective even under spectral suppression For example, when perturbations are generated on CIFAR-100 and migrated to CIFAR-10, FUSE reduces accuracy to 9.98% without filtering, while baseline methods such as TUE and GUE exceed 90% This indicates that the generality of FUSE surpasses the limitations of existing methods.

**Weaknesses:**

1. Although FUSE aims to distribute perturbations across the entire frequency range, the paper mentions that when perturbations only act on high-frequency regions, low-frequency structures (such as object shapes) still have information and continue to support learning when discussing existing UE methods. Although FUSE attempts to address this issue through the CBG mechanism, its complete non learnability for extremely low-frequency semantic information still requires further theoretical analysis and experimental verification to ensure that low-frequency information is not exploited by the model in extreme situations.
2. The paper points out that larger perturbations ($\epsilon$ values) will inevitably reduce visual imperceptibility, highlighting the inherent trade-off between perturbation intensity and image quality. Although FUSE set a perturbation budget of $\epsilon=8/255$ in the experiment to ensure imperceptibility, the tolerance for perturbation intensity and image quality may vary for different application scenarios. Although the paper mentions this trade-off, it does not delve into how to dynamically adjust the $\epsilon$ value according to specific application needs, as well as strategies to maximize non learnability while maintaining imperceptibility.
3. In ablation research, the paper found that the frequency segmentation radius rc is a key factor affecting the performance of FUSE. Being too small (rc=0.2) can lead to insufficient low-frequency semantics, unstable guidance signals, and weak learnability; If it is too large (rc=0.8), it will excessively reduce the difference between high and low frequency bands and weaken cross band interaction. Although the paper experimentally determined that rc=0.5 is the optimal value, the optimal rc value may vary for different types of datasets or tasks. The paper can further explore how to adaptively determine the optimal RC to improve the universality and usability of FUSE.
4.  The FUSE framework introduces complex mechanisms such as random spectral masking and cross band guidance, which may increase the training time and computational resource requirements of the model. Although the paper mentions the use of NVIDIA L40S GPU for training, it does not provide a detailed discussion on the training efficiency, inference speed, and computational cost of FUSE under different hardware conditions compared to existing methods. In practical applications, computational efficiency is an important indicator for measuring the practicality of methods, and this information can be further supplemented.

**Questions:**

Please refer weaknesses.

---

> ### Author Response · Authors · 2025-11-23
>
> We thank the reviewer for the positive and constructive assessment of our work. Below, we respond to your questions and concerns in the weaknesses section.
>
> ## **Response to Weaknesses 1:**
>
> We thank the reviewer for raising this point. In the original manuscript, we evaluated the unlearnability of FUSE under various low-pass filter cutoffs (0.2~1.0) in **Figure 3** and **Figure 5**, covering the regime where meaningful semantic information is still preserved.
>
> To directly address the reviewer’s concern, we further include the **extreme low-pass cutoff of 0.1** in the revised manuscript (**Section E.2**). At this cutoff, the images become structureless and contain little semantic content, which is why this setting was not part of the main experiments.
>
> We additionally report the quantitative results for this case in **Table R4-1**. The accuracy of clean images drops significantly, confirming that the preserved ultra-low-frequency information is insufficient for normal classification. These results also show **FUSE preserves unlearnability** even under very aggressive low-pass filtering.
>
> **Table R4-1.** Test accuracy (%) under 0.1 low-pass cutoff.
>
> |           | CIFAR-10 | CIFAR-100 | SVHN  |
> | :-------- | :------: | :-------: | :---: |
> | **Clean** |  41.71   |   18.07   | 15.70 |
> | **FUSE**  |  **10.91**   |   **7.66**    | **8.68**  |
>
> ## **Response to Weaknesses 2:**
>
> In the original submission, we followed prior work on unlearnable examples (EMN [1], TUE [2], GUE [3]), all of which adopt $\epsilon=8/255$ as the common perturbation budget to ensure imperceptibility while preserving sufficient perturbation strength.
>
> We agree with the reviewer that there exists an inherent trade-off between perturbation strength and visual imperceptibility. This trade-off is partially subjective and depends on the tolerance of specific downstream applications. To quantify this relationship, we further evaluate FUSE under different perturbation bounds $\epsilon \in$ {2/255, 4/255, 8/255, 16/255, 32/255}. We have added this experiment to the revised paper (**Section D.2**). Small perturbations remain visually indistinguishable but provide weaker unlearnable effects. Larger perturbations produce stronger suppression of learnability but also lead to noticeable visual degradation.
>
> We appreciate the reviewer’s suggestion regarding dynamically adjusting $\epsilon$. While existing UE methods generally adopt a fixed perturbation budget, incorporating adaptive perturbation scaling is an interesting future direction. However, since the best value of $\epsilon$ depends on the **user's subjective preference**, it is difficult to establish a consistent criterion that a model could rely on to determine how $\epsilon$ should be adjusted.
>
> [1] Unlearnable Examples: Making Personal Data Unexploitable. Huang et al. ICLR 2021.
>
> [2] Transferable Unlearnable Examples. Ren et al.  ICLR 2023.
>
> [3] Game-Theoretic Unlearnable Example Generator. Liu et al. AAAI 2024.

---

> ### Author Response · Authors · 2025-11-23
>
> ## **Response to Weaknesses 3:**
>
> To provide a more comprehensive analysis, we additionally performed an ablation study of $r_c$ under a 0.5 low-pass filter cutoff on four datasets of different resolutions and characteristics: CIFAR-10, CIFAR-100, SVHN, and ImageNet\* (the first 100 classes of ImageNet). As shown in **Table R4-2**, the optimal $r_c$ consistently appears **near 0.5 across all datasets**, and the performance variation within this range is small. This indicates that the choice of $r_c$ is stable across datasets and not dataset-specific.
>
> **Table R4-2.** Ablation on the frequency split radius $r_c$ across four datasets.
>
> | $r_c$ | CIFAR-10  | CIFAR-100 |   SVHN    | ImageNet\* |
> | :---: | :-------: | :-------: | :-------: | :-------: |
> |  0.2  |   10.93   |   5.03    |   11.15   |   4.52    |
> |  0.3  |   10.84   |   4.97    |   11.01   |   4.16    |
> |  0.4  |   10.32   |   4.78    | **10.17** |   3.56    |
> |  0.5  | **10.13** | **4.45**  |   10.98   | **3.20**  |
> |  0.6  |   10.38   |   4.46    |   11.37   |   3.94    |
> |  0.7  |   11.43   |   5.65    |   16.57   |   5.15    |
> |  0.8  |   14.70   |   6.24    |   18.57   |   6.24    |
>
> We appreciate the reviewer’s suggestion regarding adaptive selection of $r_c$. Furthermore, we applied Bayesian Optimization to automatically search for the optimal $r_c$ on each dataset (**Table R4-3**). The automatically identified values also converge near 0.5, and the resulting accuracies differ from the fixed setting $r_c = 0.5$ by less than 0.1%–0.2%. These results confirm that the manually chosen $r_c=0.5$ is close to the global optimum.
>
> **Table R4-3.** Optimal $r_c$ identified by Bayesian Optimization and the corresponding unlearnable accuracy.
>
> |  | CIFAR-10  | CIFAR-100 |   SVHN    | ImageNet\* |
> | :---: | :-------: | :-------: | :-------: | :-------: |
> |  BO-selected $r_c$  |   0.47   |   0.51   |   0.42   |   0.46   |
> |  Test Acc (%)  |   10.06   |   4.36   |   10.01   |   3.12   |
>
> While selecting the dataset-wise optimal $r_c$ yields improvements, the overall performance differences are very small, demonstrating that FUSE is not sensitive to this hyperparameter. For simplicity and reproducibility, we therefore adopt $r_c = 0.5$ as a robust default choice that generalizes well across datasets.
>
> All additional experiments have been included in the revised paper (**Section D.6**).
>
> ## **Response to Weaknesses 4:**
>
> We thank the reviewer for the thoughtful comments. We have now revised the manuscript by carefully addressing your comments (**Section D.10**), and the details are presented in **Table R4-4**.
>
> Most existing UE methods directly store and add perturbations without any learnable generator. Therefore, for a fair comparison, we evaluate computational efficiency against the generator-based baseline GUE.  Specifically, we feed each generator with a total of 50000 images from CIFAR-10 using a batch size of 512 to measure training time, inference time, GFLOPs, and parameters. These results show that **FUSE contains fewer parameters and achieves lower inference time and computational cost**, highlighting its efficiency and practicality for real-world deployment.
>
> **Table R4-4.** Comparison of computational efficiency between GUE and FUSE.
>
> | Method   | Training time (s/epoch) | Inference time (ms / img) |   GFLOPs    | Parameters (M) |
> | :------- | :---------------------: | :-----------------------: | :---------: | :------------: |
> | **GUE**  |         321.10          |           0.194           |   3502.07   |      7.79      |
> | **FUSE** |        **21.24**        |         **0.004**         | **1146.28** |    **0.09**    |

---

> > ### Comment · Reviewer_Vrst · 2025-11-25
> >
> > Dear Authors,
> >
> > I thank the authors for their detailed response.
> >
> > I have also carefully read the comments from the other reviewers. At this moment, I will maintain my current rating and make final decision after a discussion with other reviewers.
> >
> > Reviewer Vrst

---

> > > ### Author Response · Authors · 2025-11-25
> > >
> > > Dear Reviewer Vrst,
> > >
> > > We sincerely appreciate your support, thoughtful feedback, and the time you took to review our rebuttal. Please let us know if you have any additional questions or suggestions that might help clarify our work and contribute to a more informed and positive assessment. We would be glad to provide any further details that may assist during the discussion phase.
> > >
> > > Thank you again for your valuable consideration.
> > >
> > > Best regards,
> > >
> > > Submission 2708 Authors

---

### Official Review · Reviewer_DnDi · 2025-10-28

**Soundness:** 3
**Presentation:** 3
**Contribution:** 3
**Rating:** 6
**Confidence:** 4

**Summary:**

This paper proposes FUSE (Full-spectrum Unlearnable Examples), a novel framework that aims to generate spectrum-agnostic unlearnable perturbations through spectral equalization. The key idea is to distribute unlearnability across the entire frequency spectrum by combining two complementary components: Random Spectrum Masking (RSM) to enhance frequency diversity and Cross-band Guidance (CBG) to enforce consistency between different frequency bands. Experimental results on multiple datasets and architectures demonstrate that FUSE achieves strong protection against unauthorized model training, outperforming prior unlearnable example methods under low-pass filtering attacks.

**Strengths:**

1. The paper starts from a simple yet insightful observation about the vulnerability of existing unlearnable examples to frequency filtering, and develops a very intuitive and targeted idea to overcome it.
2. The writing is fluent, clear, and well-organized, making the paper easy and pleasant to read.
3. The experimental results are strong, generally reaching or surpassing state-of-the-art unlearnable example baselines under comparable settings.

**Weaknesses:**

1. Overly favorable threat model / limited generalization: FUSE is trained jointly with a surrogate classifier that is also used as the target model during evaluation. As a result, the perturbations are tightly coupled to the specific architecture used in training. This effectively assumes that the data owner knows the attacker’s model architecture—a strong and unrealistic assumption. While the paper claims cross-architecture robustness, all reported results use the same backbone for both surrogate and target models, which makes the transferability and real-world applicability of FUSE questionable.
2. Claim–evidence mismatch for “full-spectrum” robustness: The central claim of being “spectrum-agnostic” or “full-spectrum” is not convincingly supported by experiments. All frequency-domain evaluations focus solely on low-pass filtering (removing high-frequency components). There is no evidence that FUSE remains effective under high-pass or band-stop filtering, which would be necessary to substantiate the “full-spectrum” claim.
3. The current objective is written as $\min_\theta \min_\delta \mathcal{L}(\theta,\delta)$, following prior work “Unlearnable Examples: Making Personal Data Unexploitable”, ICLR 2021, which uses PGD to explicitly update $\delta$. However, since FUSE updates $\delta$ in a implicit way by using a trainable perturbation generator $\delta=\mathcal{G}(x)$, it is better to define the parameters of $\mathcal{G}$ as $\psi$ and update the objective formulation to $\min_\theta \min_\psi \mathcal{L}(\theta,\mathcal{G}_{\psi}(x))$ or clarity and correctness.
4. Given the relatively complex two-branch training and frequency masking pipeline, a concise pseudocode block would significantly improve readability.
5. It would be helpful to cite related papers, such as *DAT: Improving Adversarial Robustness via Generative Amplitude Mix-up in Frequency Domain*, NeurIPS 2024, which shares conceptual connections to frequency-domain data manipulation.

**Questions:**

See Weaknesses. I am willing to raise the score if the authors address my concerns.

---

> ### Author Response · Authors · 2025-11-23
>
> We thank the reviewer for the constructive feedback and for highlighting both the strengths and open questions in our work. Below, we show our response to your concerns in the weaknesses section. We are happy to provide additional clarification if needed.
>
> ## **Response to Weaknesses 1:**
>
> We have added a new experiment where UEs are **generated using ResNet-18** as the surrogate model but **tested on four unseen architectures**: ResNet-50, VGG-11, DenseNet-121, and ViT. All results are evaluated on CIFAR-10. The results (**Table R3-1**) show that FUSE maintains **the strongest unlearnability** across all unseen architectures, outperforming other UE methods in every case. This confirms that FUSE generalizes well and its effectiveness is not limited to a single model.
>
> We have incorporated this new cross-architecture evaluation into the revised manuscript (**Section 4.4**).
>
> **Table R3-1.** Test Cross-Architecture accuracy (%) on CIFAR-10.
>
> | Method   | ResNet-50 |  VGG-11   | DenseNet-121 |    ViT    |
> | :------- | :-------: | :-------: | :----------: | :-------: |
> | **EMN**  |   17.90   |   29.30   |    18.60     |   24.37   |
> | **TUE**  |   13.41   |   26.29   |    17.39     |   26.29   |
> | **GUE**  |   12.97   |   13.72   |    13.71     |   16.77   |
> | **PUE**  |   12.57   |   27.71   |    14.04     |   20.07   |
> | **FUSE** | **11.39** | **11.67** |  **11.95**   | **10.87** |
>
>
>
> ## **Response to Weaknesses 2:**
>
> We appreciate the reviewer’s insightful comment about the need to evaluate high-frequency robustness. To address this issue, we further evaluate FUSE under high-pass filtering, which removes low-frequency components.
>
> As shown in **Table R3-2**, FUSE consistently maintains **near random-guess accuracy** across all cutoff values, demonstrating unlearnability even when low-frequency information is removed. These results complement our low-pass evaluations and together provide strong evidence that FUSE is robust to the removal of **both low-frequency and high-frequency** components, thereby supporting our **spectrum-agnostic claim**. The corresponding results and analysis have been added to the revised manuscript (**Appendix D.8**).
>
> Notably, the observation that prior UE methods also remain effective under high-pass filtering is consistent with our analysis in the main paper: effective perturbation signals for unlearnability concentrate predominantly in high frequencies.
>
> **Table R3-2.** Unlearnability performance under high-pass filtering with different cutoffs.
>
> | High-frequency cutoff |   0.1    |    0.2    |    0.3    |    0.4    |   0.5    |   0.6    |   0.7    |    0.8    |    0.9    |
> | :-------------------- | :------: | :-------: | :-------: | :-------: | :------: | :------: | :------: | :-------: | :-------: |
> | **Clean**             |  94.69   |   94.43   |   93.65   |   93.45   |  92.30   |  91.56   |  91.01   |   90.54   |   90.41   |
> | **EMN**               |  13.71   |   11.98   |   11.14   |   10.41   |  10.46   |  10.04   |  10.91   |   11.03   |   14.12   |
> | **LSP**               |  13.47   |   12.04   |   11.81   |   10.85   |   9.81   |  11.53   |  15.55   |   19.54   |   23.93   |
> | **TUE**               |  10.24   |   10.51   |   10.74   |   10.55   | **9.55** |  10.25   |  11.23   |   13.41   |   15.38   |
> | **GUE**               |  13.40   |   13.76   |   12.73   |   16.01   |  10.47   |  10.03   |  10.22   |   10.13   |   12.73   |
> | **PUE**               |  10.92   |   10.46   |   11.07   |   10.45   |  10.03   | **9.66** |  11.49   |   13.57   |   11.92   |
> | **FUSE**              | **9.41** | **10.34** | **10.68** | **10.24** |  10.37   |  10.00   | **9.91** | **10.05** | **11.39** |

---

> ### Author Response · Authors · 2025-11-23
>
> ## **Response to Weaknesses 3:**
> We thank the reviewer for pointing out the inconsistency in the objective formulation. In the revised manuscript, we now explicitly treat the perturbation as $\delta = G_\psi(x)$, where $\psi$ denotes the learnable parameters of the generator (**Section 3.1**).
>
> Accordingly, the optimization objective has been corrected to $\min_{\theta}\min_{\psi}\mathcal{L}\big(\theta,\, G_\psi(x)\big)$. This updated formulation avoids interpreting $\delta$ as an independent variable (**Section 3.4**). This updated formulation accurately reflects the actual training procedure and resolves the ambiguity in the previous version.
>
> ## **Response to Weaknesses 4:**
> Following the reviewer’s suggestion, we have added a pseudocode block in the revised version (**Section 3.4**), summarizing the training pipeline, the randomized spectral masking, and the cross-band guidance. We believe this addition greatly improves the clarity and reproducibility of the method.
>
> ## **Response to Weaknesses 5:**
> We appreciate the pointer to this related work. We have included a citation to *DAT: Improving Adversarial Robustness via Generative Amplitude Mix-up in Frequency Domain* in the revised manuscript and added a brief discussion highlighting its conceptual relevance to frequency-domain perturbation designs (**Section 2**). Thank you for bringing this connection to our attention.

---

> > ### Comment · Reviewer_DnDi · 2025-11-26
> >
> > Thank you for the detailed response. Most of my earlier concerns have been addressed, and I think the paper now meets the ICLR acceptance bar. I also note that Reviewer KZCx has raised several issues that remain unresolved. I will discuss with other reviewers and the AC before making a final decision on whether to raise my score. Good luck!

---

> > > ### Author Response · Authors · 2025-11-27
> > >
> > > We sincerely thank Reviewer DnDi for the recognition of our work and for the positive response. We are glad to see that your earlier concerns have been addressed. If you have any further questions or suggestions, please feel free to let us know, and we will be happy to provide additional clarification or details to assist in the discussion phase. Thank you once again for your time and for considering our work.

---

### Official Review · Reviewer_eftW · 2025-10-30

**Soundness:** 4
**Presentation:** 2
**Contribution:** 3
**Rating:** 6
**Confidence:** 4

**Summary:**

The paper observes that existing methods for generating unlearnable examples can be defeated by applying a low-pass filter. They then proceed to defeat this defence by making unlearnable examples that work across all frequency bands.

**Strengths:**

* The introduction and motivation of the method is clear.
* Figure 1 gives clear evidence that a low-pass filter defeats other methods.
* While the method is complex, the ablation experiments show that all parts of the method are needed.
* Source code is available.

**Weaknesses:**

* The experimental setting is not clearly specified: is this a class-wise or sample-wise attack? How much of the data is attacked?
* The architecture is also not made clear enough. Are you generating individual unlearnable examples, or training a generator that turns any input into an unlearnable example? And in the second case, what is the architecture of the generator, and what data is used to train this generator?
* Only the optimization objective is presented, not the optimization method.
* Because these details are missing, it would be hard to reproduce the results from only the paper.
* The method was only tested on small images (32×32 pixels), parameters like spectral cutoff might need to be very different for larger images.

**Questions:**

* Is it necessary to use structural similarity in the cross-band-guidance term? I would expect that only cosine similarity would be enough.
* "Formally, for a perturbation δ, its shifted Fourier transform is defined as $\hat{\delta} = \text{fftshift}(F[\delta])$."
  This is not a useful definition, because it doesn't say what fftshift is.

---

> ### Author Response · Authors · 2025-11-23
>
> We greatly appreciate your constructive and detailed feedback! Below are our responses to your questions and concerns in the weaknesses part. We hope this will help you understand our paper better. We are happy to provide additional clarification if needed.
>
> ## **Response to Weaknesses 1:**
>
> Our method is a **class-wise** UE method, and we add perturbations to the **entire training dataset**, which is the common practice in UE literature, to make all user data unexploitable by unauthorized training. This full-set perturbation setting is adopted consistently across prior works such as EMN [1], TUE [2], and GUE [3]. Our experimental setting follows the same standard protocol. We have explicitly clarified this setting in the revised manuscript (**Appendix C**).
>
> To further address the reviewer’s concern regarding how much of the data is attacked, we additionally follow the analysis of **Different Unlearnable Percentages** [1] [4]. Specifically, for each unlearnable percentage, we train two models. The first model uses both the clean subset and the unlearnable subset as its training data, and the second one only uses the clean subset. The difference between the performances of those two models represents how much semantic information the first model gains from the UEs. A small performance gap indicates the first model gains little information from the UEs. As shown in **Table R2-1** and **Table R2-2**, all experiments are conducted on CIFAR-10. To avoid ambiguity, **Table R2-1** reports the unlearnability results under the low-pass filtered evaluation with a cutoff radius of 0.5, whereas **Table R2-2** presents the corresponding results without any filtering. FUSE consistently achieves the best and second-best performance across all settings.
>
> All additional experiments have been included in the revised paper (**Appendix D.4**).
>
> **Table R2-1.** Test accuracy (%) for different unlearnable percentages $p$ under a low-pass filtering operation.
>
> | Percentages $p$ |    10%    |    20%    |    30%    |    40%    |    50%    |    60%    |    70%    |    80%    |    90%    |    100%    |
> | :---------------- | :-------: | :-------: | :-------: | :-------: | :-------: | :-------: | :-------: | :-------: | :-------: | :-------: |
> | Clean (1- $p$)  |   85.39   |   85.26   |   84.23   |   83.70   |   83.01   |   82.24   |   81.13   |   77.30   |   75.49   |     -     |
> | EMN               |   90.53   |   87.55   |   87.13   |   86.50   |   85.72   |   85.22   | 84.95 |   83.93   |   83.28   |   23.56   |
> | LSP               |   90.30   |   88.90   |   88.52   |   87.93   |   87.81   |   87.05   |   86.58   |   85.96   |   85.32   |   54.16   |
> | TUE               |   90.04   |   87.78   |   88.35   |   87.82   |   87.59   |   87.58   |   87.43   |   86.79   |   85.03   |   52.16   |
> | GUE               |   90.01   |   89.24   |   89.04   |   88.71   |   88.14   |   87.62   |   87.28   |   86.72   | 85.78 |   41.97   |
> | PUE               | 89.51 |   88.97   |   88.54   |   88.22   |   87.88   |   87.40   |   87.26   |   86.72   |   84.04   |   23.60   |
> | **FUSE**          |   **87.29**   | **86.41** | **85.92** | **85.06** | **84.94** | **84.13** |   **83.12**   | **79.38** |   **77.76**   | **10.13** |
>
> **Table R2-2.** Test accuracy (%) for different unlearnable percentages $p$ without any filtering applied.
>
> | Percentages $p$ |    10%    |    20%    |    30%    |    40%    |    50%    |    60%    |    70%    |    80%    |    90%    |    100%    |
> | :---------------- | :-------: | :-------: | :-------: | :-------: | :-------: | :-------: | :-------: | :-------: | :-------: | :-------: |
> | Clean (1- $p$)   |   94.01   |   93.75   |   93.10   |   92.56   |   92.42   |   89.77   |   86.44   |   84.30   |   82.64   |     -     |
> | EMN               |   95.29   |   94.24   | 93.88 |   92.99   |   92.87   |   91.10   |   88.52   |   87.23   |   85.21   |   16.42   |
> | LSP               |   95.70   |   **94.04**   |   93.79   |   93.10   |   92.91   |   91.96   |   88.93   |   86.29   |   85.73   |  13.54   |
> | TUE               |   95.18   |   94.84   |   94.35   |   93.88   |   92.68   |   91.05   |   88.19   |   86.62   |   85.39   |   10.03   |
> | GUE               |   95.78   |   95.07   |   94.48   |   93.94   |   93.15   |   90.41   |   85.40   | **83.99** |   83.12   |   13.25   |
> | PUE               |   95.10   |   94.84   |   94.30   |   93.69   |   93.24   |   91.74   |   88.75   |   86.10   |   85.07   |   10.62   |
> | **FUSE**          |   **94.79**   | 94.21 | **93.64** | **92.37** | **92.28** | **90.11** |   **85.38**   | 84.66 |   **82.93**   | **9.41** |
>
> [1] Unlearnable Examples: Making Personal Data Unexploitable. Huang et al. ICLR 2021.
>
> [2] Transferable Unlearnable Examples. Ren et al.  ICLR 2023.
>
> [3] Game-Theoretic Unlearnable Example Generator. Liu et al. AAAI 2024.
>
> [4] Availability Attacks Create Shortcuts. Yu et al. SIGKDD 2022.

---

> ### Author Response · Authors · 2025-11-23
>
> ## **Response to Weaknesses 2:**
>
> We appreciate the reviewer’s question regarding the architecture and training procedure of our generator.
>
> As described in **Section 3.1**, we train a generator network that takes an input image $x$ and outputs a perturbation $\delta = G_\psi(x)$, which is then added to the image to produce the unlearnable example $x' = x + \delta$.  The architecture of the generator is described in **Appendix B**, including down-sampling layers, residual blocks, and up-sampling layers.
>
> ## **Response to Weaknesses 3:**
>
> The optimization of FUSE follows an **alternating min-min optimization scheme**. During the first minimization loop, the surrogate classifier $f_\theta$ is optimized to minimize the classification loss, while the generator $G_\psi$ is kept fixed. During the second minimization loop, the perturbation generator $G_\psi$ is optimized to also minimize the RSM and CBG loss, while $f_\theta$ is kept fixed. We add a **pseudocode block** in the revised version (**Section 3.4**) to clarify the optimization method.
>
> ## **Response to Weaknesses 4:**
>
> To address the reviewer’s concern regarding reproducibility, we have added all missing implementation details to the revised manuscript.
> Specifically, we now clearly describe (1) the *attack setting* which includes class-wise method (**Appendix C**), full-dataset perturbation (**Appendix C**), and additional experiments using different unlearnable percentages (**Appendix D.4**), (2) the *generator architecture* and the data used to train it (**Section 3.1** and **Appendix B**), and (3) the *optimization procedure*, including the full alternating min–min training algorithm with pseudocode (**Section 3.4**).
>
> With these additions, the revised paper contains all necessary details, making the entire FUSE pipeline reproducible.
>
> ## **Response to Weaknesses 5:**
>
> To investigate how the spectral cutoff parameter scales with image resolution, we conducted a sensitivity analysis of the frequency split radius $r_c$ on ImageNet\* (the first 100 classes of ImageNet). These experiments were evaluated using test accuracy under a 0.5-cutoff low-pass filter. The results are summarized in **Table R2-3**, and are further discussed in **Appendix D.6**. The findings show that the performance remains stable across a broad range, even under high-resolution settings.
>
> **Table R2-3.** Unlearnability comparisons under different values of $r_c$.
>
> | $r_c$     | 0.2  | 0.3  | 0.4  |   0.5    | 0.6  | 0.7  | 0.8  |
> | --------- | :--: | :--: | :--: | :------: | :--: | :--: | :--: |
> | ImageNet\* | 4.52 | 4.16 | 3.56 | **3.20** | 3.94 | 5.15 | 6.24 |
>
> ## **Response to Question 1:**
>
> We thank the reviewer for the insightful question. We agree that cosine similarity alone is sufficient for enforcing directional consistency between the low- and high-frequency components. However, we include the structural similarity (SSIM) term in the cross-band-guidance module for a different purpose: **to preserve spatial and perceptual cues**, rather than to improve feature-level similarity.
>
> To clarify this distinction, we conduct an ablation study by removing the SSIM term while keeping all other components unchanged, and evaluate both variants on CIFAR-10 without any frequency filtering. As shown in **Table R2-4**, removing the structural term **degrades image quality** across multiple perceptual metrics. We have added this clarification to the revised manuscript (**Section D.9**).
>
> **Table R2-4.** Effect of structural similarity.
>
> | Method        | PSNR $\uparrow$ | SSIM $\uparrow$ | LPIPS $\downarrow$ | MSE $\downarrow$ |
> | :------------ | :-------------: | :-------------: | :----------------: | :--------------: |
> | w/o structure |      37.17      |     0.9707      |       0.0374       |     0.000358     |
> | FUSE          |    **39.35**    |   **0.9923**    |     **0.0163**     |   **0.000166**   |
>
> ## **Response to Question 2:**
>
> Thank you for the question. We have clarified in the revision that $\mathcal{F}[\delta]$ denotes the 2D FFT of the perturbation. The term “shifted” refers to the standard centered Fourier representation, where the zero-frequency component is placed at the center of the spectrum rather than at the corner. In implementation, this corresponds to applying *fftshift* after the FFT. This clarification has been added near Eq. (3) (**Section 3.2**).

---

### Official Review · Reviewer_KZCx · 2025-11-03

**Soundness:** 2
**Presentation:** 3
**Contribution:** 2
**Rating:** 4
**Confidence:** 5

**Summary:**

The paper identifies a major weakness in current Unlearnable Examples (UEs): their protective perturbations are concentrated in high frequencies and can be easily broken by low-pass filtering. To address this, the authors propose FUSE, a new framework for generating "full-spectrum" UEs. FUSE uses two main strategies: 1) Random Spectral Masking, which strengthens the perturbations across all frequencies by forcing the system to work even when a contiguous band is removed, and 2) Cross-Band Guidance, which aligns the high and low-frequency components to enhance low-frequency effectiveness and maintain the original image's visual fidelity.

**Strengths:**

The paper is well-structured and clearly presented. It demonstrates sufficient novelty, and the core idea of coupling unlearnability across both the low-frequency and high-frequency bands is compelling.

**Weaknesses:**

**Ambiguity**:
+ 1. In Equation (3), the definition of the shifted Fourier transform is somewhat unclear. What does $F(\delta)$ represent? Specifically, what is being "shifted" in this context?

+ 2. The details regarding the cutoff require further elaboration. Equation (6) does not provide useful information on how the cutoff is determined or implemented.

+ 3. The authors should provide a more thorough explanation of how loss function (7) achieves the goal of transferring the information that makes the model less learnable from high frequencies to low frequencies. The current explanation does not sufficiently connect the design of the loss to the stated purpose above.

**Experiment Design**
+ 1. A critical omission from the experimental evaluation is the analysis of generalization. Assessing the transferability of unlearnable examples (UEs) to unknown target models is a vital aspect of their evaluation. The current results lack this dimension. The authors state in the experimental setup: "We use ResNet-18..., as the surrogate classifier and target model both in training and testing." However, the subsequent results (e.g., in Table 1) exclusively report performance where the surrogate and target models are identical. This does not constitute a generalization test. The key experiment—evaluating UEs trained on one surrogate model against a different, unseen target model—is missing and its absence severely undermines the effectiveness of FUSE.

+  2. Section 4.3 discusses the impact of different $r_c$ values. In this set of experiments, is the $r_c$ used for training the UE kept fixed, or does it vary along with the $r_c$ of the low-pass filter used for defense? In other words, does FUSE also exhibit generalization capability with respect to $r_c$? After all, in practical scenarios, it is unknown what $r_c$ value a defender might adopt for the low-pass filter.


+ 3. Regarding potential defense strategies: JPEG compression is a very common preprocessing method in real-world scenarios. Since JPEG typically has a significant impact on the frequency-domain information of UEs, and FUSE incorporates substantial design focused on the frequency domain, I suspect that JPEG compression could considerably undermine the effectiveness of FUSE, substantially weakening its unlearnability capability.

+ 4. The current image resolution under consideration is relatively low. Thus, the range of the frequency band is also quite limited. When the image size increases, the range of the frequency band expands accordingly. At this point, the parameters of the algorithm, such as $r_c$, may become more sensitive. Additionally, it is no longer sufficient to simply categorize the information as high-frequency or low-frequency, as it may also include mid-to-high frequency, mid-to-low frequency, and other components. Therefore, the lack of experiments in this area makes it difficult to fully validate the effectiveness of the method.

**Questions:**

See the weaknesses.

---

> ### Author Response · Authors · 2025-11-23
>
> We appreciate your detailed feedback. Below, we address the concerns and questions raised in the weaknesses section. Please feel free to reach out if further clarification is required.
>
> ## **Response to Ambiguity 1:**
> Thank you for the question. We have clarified in the revision that $\mathcal{F}[\delta]$ denotes the 2D FFT of the perturbation. The term “shifted” refers to the standard centered Fourier representation, where the zero-frequency component is placed at the center of the spectrum rather than at the corner. In implementation, this corresponds to applying *fftshift* after the FFT. This clarification has been added near Eq. (3) (**Section 3.2**).
>
> ## **Response to Ambiguity 2:**
>
> Thank you for pointing this out. The frequency split radius $r_c$ is a hyperparameter and is not learned. We clarify the exact formulation below and include this clarification in the revised manuscript (**Section 3.3**).
>
> In our method, the cutoff radius denotes the normalized spectral radius, i.e., the proportion of the Fourier spectrum retained as low-frequency components. For an image of spatial size $(H \times W)$, we compute the normalized radial frequency for each location $(i,j)$ in the Fourier domain:
>
> $
> r(i,j) =\sqrt{\left(\frac{i - H/2}{H/2}\right)^2 +
> \left(\frac{j - W/2}{W/2}\right)^2
> }.
> $
>
> The low-frequency and high-frequency masks are binary circular masks defined as:
>
> $M_{\text{low}}(i,j) = \mathbf{1}[ r(i,j) \le r_c ], \qquad
> M_{\text{high}} = 1 - M_{\text{low}}.$
>
> We apply these masks directly to the Fourier transform of the input:
>
> $x_{\text{low}}
> = \mathcal{F}^{-1}( M_{\text{low}} \odot \mathcal{F}(x) ), \qquad
> x_{\text{high}}
> = \mathcal{F}^{-1}( M_{\text{high}} \odot \mathcal{F}(x) ).$
>
> This normalized, resolution-independent definition ensures that the cutoff has the same semantic meaning across different image sizes.
>
> ## **Response to Ambiguity 3:**
>
> Thank you for the comment. In this revision we have carefully addressed this aspect for greater clarity and add this explanation to the revised version (**Section 3.3**).
>
> High-frequency UEs naturally induce stronger unlearnability. Formally, the output $\hat y\_{\text{high}} = f_\theta(x'\_{\text{high}})$ already encodes **a distribution that reflects the model’s confusion under high-frequency perturbations**. In contrast, low-frequency UEs carry stronger semantic content and therefore tend to remain more learnable unless explicitly constrained. Eq. (7) establishes this constraint:
>
> $
> \mathcal{L}\_{\text{guide}}
> = \mathcal{L}\_{CE}\big(f_\theta(x'\_{\text{low}}), \hat y\_{\text{high}}\big).
> $
>
> By minimizing this loss, the low-frequency pathway is required to **imitate the predictive behavior produced by the high-frequency perturbation**. Since $\hat y_{\text{high}}$ contains the unlearnability signal, this process effectively distills the high-frequency confusion into the low-frequency pathway so that the model cannot circumvent unlearnability by relying on frequency-based shortcuts.
>
> ## **Response to Experiment Design 1:**
>
> We have added a new experiment where UEs are **generated using ResNet-18** as the surrogate model but **tested on four unseen architectures**: ResNet-50, VGG-11, DenseNet-121, and ViT. All results are evaluated on CIFAR-10. The results (**Table R1-1**) show that FUSE maintains **the strongest unlearnability** across all unseen architectures, outperforming other UE methods in every case. This confirms that FUSE generalizes well and its effectiveness is not limited to a single model.
>
> We have incorporated this new cross-architecture evaluation into the revised manuscript (**Section 4.4**).
>
> **Table R1-1.** Test Cross-Architecture accuracy (%) on CIFAR-10.
>
> | Method   | ResNet-50 |  VGG-11   | DenseNet-121 |    ViT    |
> | :------- | :-------: | :-------: | :----------: | :-------: |
> | **EMN**  |   17.90   |   29.30   |    18.60     |   24.37   |
> | **TUE**  |   13.41   |   26.29   |    17.39     |   26.29   |
> | **GUE**  |   12.97   |   13.72   |    13.71     |   16.77   |
> | **PUE**  |   12.57   |   27.71   |    14.04     |   20.07   |
> | **FUSE** | **11.39** | **11.67** |  **11.95**   | **10.87** |
>
> ## **Response to Experiment Design 2:**
> As stated in **Sec. 4.1 (Implementation Details)**, we fix the frequency split radius in Eq. 6 to  $r_c = 0.5$ throughout all experiments.  This radius is only used to decompose the perturbation into low- and high-frequency components inside FUSE. It defines the internal structure of FUSE and does not correspond to any defender operation. In other words, the model cannot revert to a clean semantic representation even if the defender applies a different low-pass filter. FUSE provides stable unlearnability to unseen defender cutoffs, as shown in **Section 4.3 and Figure 3**.

---

> ### Author Response · Authors · 2025-11-23
>
> ## **Response to Experiment Design 3:**
>
> To address the reviewer’s concern,  we tested the impact of JPEG compression on UE performance evaluated on CIFAR-10 under a low-pass filtered scenario. The results are presented in **Table R1-2**, which shows the unlearnability performance of our method under these different frequency-domain conditions, demonstrating that our method remains effective even under strong frequency-domain distortions.
>
> **Table R1-2.** Test accuracy (%) under different JPEG compression qualities.
>
> | JPEG Compression |    10     |    20     |    30     |    40     |    50     |    60     |    70     |    80     |    90     |
> | :--------------- | :-------: | :-------: | :-------: | :-------: | :-------: | :-------: | :-------: | :-------: | :-------: |
> | **Clean**        |   90.69   |   90.43   |   90.65   |   90.45   |   90.30   |   90.56   |   91.01   |   90.54   |   90.41   |
> | **EMN**          |   81.93   |   80.30   |   76.37   |   72.49   |   69.07   |   65.89   |   62.01   |   42.56   |   25.48   |
> | **LSP**          |   82.03   |   83.14   |   81.59   |   78.89   |   76.57   |   72.27   |   68.17   |   62.72   |   57.48   |
> | **TUE**          |   59.53   |   58.63   |   54.10   |   57.13   |   56.25   |   57.15   |   55.89   |   54.46   |   54.62   |
> | **GUE**          |   78.88   |   78.24   |   74.53   |   72.56   |   71.26   |   66.32   |   60.59   |   50.41   |   43.38   |
> | **PUE**          |   54.84   |   45.99   |   43.78   |   37.63   |   31.73   |   29.80   |   27.40   |   25.89   |   23.76   |
> | **FUSE**         | **54.56** | **35.16** | **31.60** | **27.32** | **25.41** | **20.77** | **18.90** | **12.87** | **10.98** |
>
>
>
> ## **Response to Experiment Design 4:**
>
> **4.1** *The current image resolution under consideration is relatively low. Thus, the range of the frequency band is also quite limited. When the image size increases, the range of the frequency band expands accordingly. At this point, the parameters of the algorithm, such as $r_c$, may become more sensitive.*
>
> Regarding this concern, we have conducted additional experiments to analyze the sensitivity of $r_c$ on ImageNet\* (the first 100 classes of ImageNet), evaluated by test accuracy under a low-pass filter with a 0.5 cutoff. The results are summarized in the **Table R1-3**, and are further discussed in **Appendix D.6**. The findings show that the performance remains stable across a broad range, even under high-resolution settings.
>
> **Table R1-3.** Unlearnability comparisons under different values of $r_c$.
>
> | $r_c$     | 0.2  | 0.3  | 0.4  |   0.5    | 0.6  | 0.7  | 0.8  |
> | --------- | :--: | :--: | :--: | :------: | :--: | :--: | :--: |
> | ImageNet* | 4.52 | 4.16 | 3.56 | **3.20** | 3.94 | 5.15 | 6.24 |
>
> **4.2** *Additionally, it is no longer sufficient to simply categorize the information as high-frequency or low-frequency, as it may also include mid-to-high frequency, mid-to-low frequency, and other components. Therefore, the lack of experiments in this area makes it difficult to fully validate the effectiveness of the method.*
>
> We fully agree that introducing more splits of frequency could further improve the performance. Specifically, following your suggestion, we conducted an additional experiment using a three-band (low, mid, high) frequency partition on ImageNet\*. We further applied our CBG module (Section 3.3) to all pairwise band interactions (low-to-mid, low-to-high, and mid-to-high). The results are summarized in **Table R1-4** and revised in **Appendix D.7**. It can be observed that, while modest, your suggestion indeed leads to stronger unlearnability. On the other hand, we note that both two-band and three-band designs achieve accuracies that are close to random guess, whereas the latter incurs higher computational cost, indicating a trade-off between effectiveness and efficiency.
>
> **Table R1-4.** Effect of the Number of Frequency Bands.
>
> | Method           | Low-pass filter | Unfiltered | Training time (s/epoch) |
> | ---------------- | :-------------: | :--------: | :---------------------: |
> | Two-band split   |      3.20       |    1.12    |        **21.24**        |
> | Three-band split |    **2.84**     |  **1.02**  |          32.03          |

---

> ### Comment · Reviewer_KZCx · 2025-11-26
> **Experiment about the impact of JPEG.**
>
> I have several concerns about the authenticity of experimental results regarding the impact of JPEG compression. First, I am confused why such strong JPEG compression (quality 10) does not affect the model’s performance on clean images at all. This seems to contradict the findings in the reference (https://arxiv.org/pdf/2006.08145) and our own experiments, where we observed a drop to 84% accuracy for ResNet18 under similar conditions.
>
> More importantly, I still believe that JPEG compression should significantly affect this method, since it relies on frequency information. The authors claim that even with JPEG quality 70, their method can successfully poison the victim model, reducing accuracy to 18%. However, in my own testing, the accuracy rose sharply to 58.30% even under JPEG 75, which suggests much weaker effectiveness.
>
> As for ImageNet, I suspect the effect of JPEG compression would be even more noticeable compared to CIFAR-10, since ImageNet images are much larger in size and may respond differently to compression.

---

> > ### Author Response · Authors · 2025-11-27
> >
> > 1. *I have several concerns about the authenticity of experimental results regarding the impact of JPEG compression. First, I am confused why such strong JPEG compression (quality 10) does not affect the model’s performance on clean images at all. This seems to contradict the findings in the reference (https://arxiv.org/pdf/2006.08145) and our own experiments, where we observed a drop to 84% accuracy for ResNet18 under similar conditions.*
> >
> > Thank you very much for carefully checking the JPEG-related results. After receiving your comment, we re-examined the evaluation pipeline and identified an implementation mistake in the “clean + JPEG” experiments: although the JPEG Compression was defined, it was not actually applied to the clean dataloader. This caused the clean accuracy under JPEG to be incorrectly reported as unchanged. We sincerely apologize for this oversight. We **have corrected the code and re-run all clean experiments**. The updated results (**Table R1-5**) follow the expected degradation trends reported in prior work and are consistent with your own observations. We will replace the incorrect numbers in the revision (Section D.7).
> >
> > Importantly, we have carefully verified that **all other JPEG Compression experiments were conducted with the correct pipeline**. In the revised supplemental material, we provide:
> > - the complete code of FUSE (including JPEG Compression),
> > - the complete experiment logs,
> > - the generator checkpoints, which were optimized on CIFAR-10 and ImageNet.
> >
> > We greatly appreciate your careful reproduction effort. We are unable to fully explain the differences in results at this point. To provide further clarification, we have shared our checkpoints and would be happy to discuss this matter. We would greatly appreciate it if you could provide more details regarding your reproduction setup.
> >
> > **Table R1-5.** Test accuracy (%) under different JPEG compression qualities on CIFAR-10.
> >
> > | JPEG Compression |    10     |    20     |    30     |    40     |    50     |    60     |    70     |    80     |    90     |
> > | :--------------- | :-------: | :-------: | :-------: | :-------: | :-------: | :-------: | :-------: | :-------: | :-------: |
> > | **Clean**        |   82.24   |   84.40   |   85.05   |   85.84   |   86.09   |   86.60   |   87.25   |   87.67   |   87.90   |

---

> > > ### Author Response · Authors · 2025-11-27
> > >
> > > 2.  *More importantly, I still believe that JPEG compression should significantly affect this method, since it relies on frequency information. The authors claim that even with JPEG quality 70, their method can successfully poison the victim model, reducing accuracy to 18%. However, in my own testing, the accuracy rose sharply to 58.30% even under JPEG 75, which suggests much weaker effectiveness.*
> > > *As for ImageNet, I suspect the effect of JPEG compression would be even more noticeable compared to CIFAR-10, since ImageNet images are much larger in size and may respond differently to compression.*
> > >
> > > We agree that  JPEG compression could considerably undermine the effectiveness of UE methods. As noted in Section 3.2 of [1], "*JPEG has to **represent the low frequencies with a high accuracy**, whereas JPEG can use **coarse measuring sticks to represent the high frequencies** without jeopardizing the visual content of the blocks.*"  This means  JPEG compression is used to **eliminate high-frequency shortcuts** [2]. However, since FUSE proposes **full-spectrum unlearable perturbations**, rather than relying on the high-frequency band perturbations like other UE methods, it is more robust to the JPEG compression compared with other UE methods.
> > >
> > > To verify this, we present additional experiments in **Section D.7**, where we show t-SNE visualizations with and without low-pass filtering after JPEG compression (Q=70) for different methods. These results are consistent with the reviewer's analysis. Specifically, under full-spectrum conditions, FUSE’s perturbations result in low accuracy, while the unlearnable performance of other methods decreases.  When low-pass filtering is applied after JPEG compression, FUSE’s performance remains **close to random-guess level**, demonstrating that **FUSE is more robust to JPEG compression than other methods**.
> > >
> > > Additionally, we test FUSE on an ImageNet subset, which we randomly select 20% images from the first 100 classes of the official ImageNet training set and validation set. As shown in **Table R1-6**, the accuracy remains relatively low across most compression settings, indicating that although JPEG partially disrupts the perturbation pattern, it does not **completely eliminate FUSE's unlearnable effect**. Due to time and computational constraints during the rebuttal period, we use only 20% of the validation set, which may introduce minor variance in the results. However, the results still demonstrate FUSE's unlearnable performance under JPEG Compression. The checkpoint and logs for ImageNet are also provided in the supplementary material.
> > >
> > > **Table R1-6.** Test accuracy (%) under different JPEG compression qualities on ImageNet subset.
> > >
> > > | JPEG Compression |  10   |  20   |  30   |  40   |  50   |  60   |  70   |  80   |  90   |
> > > | :--------------- | :---: | :---: | :---: | :---: | :---: | :---: | :---: | :---: | :---: |
> > > | **Clean**        | 54.60 | 58.30 | 58.40 | 59.30 | 59.40 | 59.80 | 60.00 | 60.60 | 62.30 |
> > > | **FUSE**   | **5.70** | **9.50** | **3.30** | **2.80** | **6.70** | **8.60** | **7.70** | **5.50** | **6.00** |
> > >
> > >
> > > [1] JPEG-1 standard 25 years: past, present, and future reasons for a success. Hudson et al. Journal of Electronic Imaging.
> > >
> > > [2] Image Shortcut Squeezing: Countering Perturbative Availability Poisons with Compression. Liu et al. ICML 2023.

---

### Author Response · Authors · 2025-12-03
**Summary of Responses and Revisions**

Dear AC/SAC/PC,

We sincerely thank the ICLR reviewers, ACs, and PCs for their time and constructive feedback. Below, we would like to provide a final summary of our contributions, discussion, and revisions.

## **Contributions and Strengths**

In this work, we identify a critical vulnerability of existing unlearnable examples, showing that **their effectiveness collapses under low-pass filtering**. To address this issue, we propose **FUSE**, a novel spectrum-agnostic framework that **distributes perturbation effects across the entire frequency range**, ensuring that unlearnability is preserved even when specific bands are suppressed by filtering.

We are encouraged that the reviewers recognized the following strengths of our method:

-  **Motivation.** Reviewers agreed that the full-spectrum perspective on unlearnable examples is novel and well-motivated, and the analysis of the limitations of narrow-band UE approaches is clear. [Reviewer `KZCx`, `eftW`, `DnDi`, `Vrst`]

-  **Method Design.** The spectrum-agnostic design of FUSE is intuitive, simple to implement, and well aligned with its underlying motivation. Reviewers also noted the frequency-based structure of the method to be reasonable and informative. [Reviewer `KZCx`, `eftW`, `DnDi`, `Vrst`]

-  **Experimental Strength.** The experiments are extensive and well-designed, and FUSE demonstrates strong performance. [Reviewer `eftW`, `DnDi`, `Vrst`]

-  **Writing & Clarity.** Reviewers highlighted that the writing is fluent, clear, and well-organized, making the paper easy and pleasant to read. [Reviewer `KZCx`, `eftW`, `DnDi`, `Vrst`]

## **Summary of the Discussion**

-  Reviewer `DnDi` indicated **“I am willing to raise the score if the authors address my concerns”** in the initial official review and stated that **“I think the paper now meets the ICLR acceptance bar”** during the discussion period.

-  Reviewer `Vrst` maintained the positive assessment and did not raise any additional concerns.

-  Reviewer `eftW` assigned a positive rating but had no response before the end of the discussion.

-  Reviewer `KZCx` raised a major question regarding the impact of JPEG compression. In response, we conducted a comprehensive empirical analysis and provided the full implementation, complete experiment logs, and generator checkpoints, which directly address the reviewer’s concerns, including *t-SNE visualizations* and *ImageNet-subset evaluations under JPEG compression*. Notably, **contrary to the** **Reviewer KZCx’s concern regarding the impact of JPEG compression, the** **empirical results further reinforce the superiority of FUSE:** its unlearnability stays markedly more robust than that of other baselines under JPEG compression, which is consistent with the full-spectrum UE behavior proposed in this work. Given that the reviewer did not respond further in the discussion stage, we entrust the AC with making the final, expert determination based on the presented clarifications and results.

## **Summary of Revision**

-  **Transferability and Robustness.** We added extensive evaluations across architectures, JPEG compression, frequency ablations, extreme filtering settings, and varying unlearnable percentages, showing that FUSE remains stable and maintains unlearnability across diverse conditions. [Reviewer `KZCx`, `eftW`, `DnDi`,` Vrst`]

-  **Computational Efficiency.** We evaluated computational efficiency compared with another generator-based baseline, which indicates that FUSE contains fewer parameters and achieves lower inference time and computational cost. [Reviewer `Vrst`]

-  **More Details on Implementation.** We provided all hyperparameter details, pseudocode, generator architecture, the full implementation, complete experiment logs, and generator checkpoints. These materials make the entire FUSE pipeline reproducible. [Reviewer `KZCx`, `eftW`, `DnDi`]

We hope this summary assists in your evaluation. Thank you once again for your time and consideration.



Best regards,

Submission 2708 Authors

---

### Meta-Review · Area_Chair_zSX4 · 2026-01-07

**Summary:**

The reviewers have commented positively on the strength of the motivation. At the same time, they have raised several substantive concerns: key ambiguities and missing implementation details render made reproduction hard; the experimental threat model initially looked overly favorable because surrogate and target architectures were the same; the “full-spectrum” claim initially lacked evidence beyond low-pass filtering; and robustness questions remained around defender cutoff generalization, higher-resolution settings / multi-band frequency structure, and practical defenses like JPEG compression. The JPEG issue is, in fact, the most contentious point: one reviewer has reported strong inconsistency with the authors’ JPEG results, casting doubt on the reliability of the JPEG claims and possibly broader robustness.

**Reviewer Concerns:**

The authors have responded by clarifying fftshift and cutoff masks, adding pseudocode and correcting the objective to reflect generator-parameter optimization. Furthermore, the authors have added cross-architecture transfer tests and high-pass filtering experiments to better support “full-spectrum” robustness and have  expanded robustness studies. These changes address many “missing experiment/clarity” complaints in principle. However, the rebuttal have not fully convinced the most skeptical reviewers because the JPEG compression results remain disputed: the authors admitted a pipeline mistake in “clean+JPEG,” but they could not explain the remaining reproduction gap where the reviewer observes far higher accuracies for FUSE under JPEG. The other two reviewers who have more positive evaluations, have deferred making their final decisions after discussions with the more critical reviewer.

**Reviewer Scores:**

Based on the rebuttals, and in the face of a lack of the usual discussion period, my understanding is that one reviewer has remained critical and the other reviewers with more positive perspectives have been paying attention to that and have deferred making their final decisions after the discussion period. Overall, my evaluation is that after the discussion period, the ratings would not have changed in a significant positive way to warrant publication.

---

### Decision · Program_Chairs · 2026-01-26

Reject